# Mapping groundwater abstractions from irrigated agriculture: big data, inverse modeling and a satellite-model fusion approach

Oliver Miguel Lopez Valencia[1], Kasper Johansen[1], Bruno José Aragon Solorio[1], Ting Li[1], Rasmus Houborg[2], Yoann Malbeteau[1], Samer AlMashharawi[1], Muhammad Umer Altaf[1], Essam Mohammed Fallatah[3], Hari Prasad Dasari[4], Ibrahim Hoteit[4] and Matthew Francis McCabe[1]

[1]Water Desalination and Reuse Center, King Abdullah University of Science and Technology, Thuwal, Saudi Arabia
[2]Planet, San Francisco, CA 94107, USA
[3]National Center for Water Research and Studies, Ministry of Environment Water and Agriculture, Riyadh, Saudi Arabia
[4]Physical Science and Engineering Division, King Abdullah University of Science and Technology, Thuwal, Saudi Arabia

*Correspondence to*: Oliver Lopez (oliver.lopez@kaust.edu.sa)

**Abstract.** The agricultural sector in Saudi Arabia has witnessed rapid growth in both production and area under cultivation over the last few decades. This has prompted some concern over the state and future availability of fossil groundwater resources, which have been used to drive this expansion. Large-scale studies using satellite gravimetric data show a declining trend over this region. However, water management agencies require much more detailed information on both the spatial distribution of agricultural fields, and their varying levels of water exploitation through time, than coarse gravimetric data can provide. Relying on self-reporting from farm operators or sporadic data collection campaigns to obtain needed information are not feasible options, nor do they allow for retrospective assessments. In this work, a water accounting framework that combines satellite data, meteorological output from weather prediction models, and a modified land surface hydrology model, was developed to provide information on both irrigated crop-water use and groundwater abstraction rates. Results from the local-scale, comprising several thousand individual center-pivot fields, were then used to quantify the regional-scale response. To do this, a semi-automated approach for the delineation of center-pivot fields using a multi-temporal statistical analysis of Landsat 8 data was developed. Next, actual crop evaporation rates were estimated using a two-source energy balance (TSEB) model driven by leaf area index, land surface temperature, and albedo inputs, all of which were derived from Landsat 8. The Community Atmosphere Biosphere Land Exchange (CABLE) model was then adapted to use satellite-based vegetation and related surface variables, and forced with a 3 km reanalysis dataset from the Weather Research and Forecasting (WRF) model. Groundwater abstraction rates were then inferred by estimating the irrigation supplied to each individual center-pivot, which was determined via an optimization approach that considered CABLE-based estimates of evaporation and TSEB-based satellite estimates. The framework was applied over two study regions in Saudi Arabia: a small-scale experimental facility of around 40 center-pivots in Al Kharj that was used for an initial evaluation, and a much larger agricultural region in Al Jawf province comprising more than 5,000 individual fields across an area exceeding 2,500 km$^2$. Total groundwater abstraction for the year 2015 in Al Jawf were estimated at approximately 5.5 billion cubic meters, far exceeding any recharge to the groundwater system and further highlighting the need for a comprehensive water management strategy. Overall, this novel data-model

fusion approach facilitates the compilation of national-scale groundwater abstractions, while also detailing field-scale information that allows both farmers and water management agencies to make informed water accounting decisions across multiple spatial and temporal scales.

## 1 Introduction

Global water consumption has increased at an unprecedented rate during the last century, with many countries turning to groundwater as either an additional or primary source of supply to meet growing agricultural and other sectoral demands (FAO 2015; Famiglietti et al., 2014). In arid and semi-arid regions in particular, groundwater is routinely the major water source driving such expansions in irrigated agriculture (Siebert et al., 2010). Unfortunately, these expansions have come with a number of associated costs related to sustainability of aquifer systems, degrading water quality and over-exploitation. Indeed,

global monitoring efforts targeting major aquifer systems around the world have identified strong depletion trends (Wada et al., 2012; Famiglietti et al., 2014), making the prospect of meeting future water and food security demands even more challenging (Dalin et al., 2017). While these relatively recent estimates of groundwater depletion (Famiglietti et al., 2011; Voss et al, 2013; Rodell et al., 2018) have been obtained through satellite systems such as the Gravity Recovery and Climate Experiment (GRACE; Tapley et al., 2004), their value as a monitoring and management tool is limited due to the coarse

observation scale (Alley and Konikow, 2015; Miro and Famiglietti, 2018). In order to provide the granularity of information needed to monitor groundwater abstractions at the field-scale (~50 ha), a combination of higher-resolution data and modeling is needed.

   Despite its extreme arid environment (Kenawy and McCabe, 2016), Saudi Arabia has quite an extensive agricultural sector. Like most national efforts to monitor and manage agricultural water use, agencies in Saudi Arabia have relied on farmer

surveys to estimate agricultural land and water extraction. In common with other national efforts, there is also a lack of regular and consistent field metering to provide measurements of agricultural water use. Nevertheless, historical estimates from early national studies indicate an agricultural extent of about 12,135 km$^2$ in 2005, with associated water use of 21 billion cubic meters (BCM) within the Kingdom (FAO 2008a). While this may be relatively small compared to other national accounts (Döll and Siebert, 2002; FAO 2008b; Wisser et al., 2008), agricultural water use in Saudi Arabia has been estimated to

represent more than 80% of the total national water consumption (FAO 2008a; Chowdhury and Al-Zahrani, 2015). Indeed, it is thought that less than 20% of the agricultural water use comes from renewable sources, with rain-fed agriculture present only in southwestern regions such as Jizan and Aseer. Local alluvial aquifers (e.g. wadis) that are occasionally recharged during storm events provide another source of water that has been used for more traditional agriculture in Saudi Arabia (Missimer et al., 2012), but these do not represent a suitable source for large commercial-scale applications. The primary origin

of water that has driven the dramatic expansion of irrigated agriculture in Saudi Arabia is non-renewable groundwater from deep fossil aquifer systems. Although agriculture has featured throughout the nation's history, significant extraction from groundwater resources really only commenced during the 1970's, when subsidies directed towards increasing food security

incentivised farmers (FAO, 2008a; Al-Rumkhani et al., 2004). These incentives, together with the relatively inexpensive access to diesel fuel required for extraction via pumping wells, combined to rapidly develop this sector of the economy. Center-pivot irrigation, where water is pumped from a well and sprayed using a rotating arm with nozzles, became the predominant method of irrigation nation-wide, and is typically applied on a daily and continuous basis for prolonged periods. The rapid expansion

in agricultural land use, especially from center-pivot irrigation fields, has thus rendered ground survey-based monitoring impractical and increasingly unreliable.

With regular coverage from Earth observing satellites, remote sensing offers a capacity to monitor the Earth across a range of spatial and temporal scales (McCabe et al., 2017a). While moderate resolution images (e.g. O~250m-1km) have been used for observing processes such as evaporation (Mu et al., 2011), these techniques lack the capacity to delineate individual

fields. Typical center-pivot fields with dimensions approaching 50 ha (800 m diameter) are generally densely vegetated, and crops with different growing seasons are often located adjacent to each other, or even within the same field. In such a case, platforms such as the MODerate resolution Imaging Spectroradiometer (MODIS) would only capture a heterogeneous mix of vegetated and bare desert soil, let alone be able to differentiate between crops (Kustas et al., 2004; McCabe et al. 2006; Wardlow et al., 2007; Guindin-Garcia et al., 2012). Landsat 8 data, on the other hand, has a spatial resolution of 30 m, allowing

it to map individual fields with a revisit time of 16 days. Whether this temporal resolution is sufficient to capture the seasonality of different crops is certainly a question to be explored, although it should be noted that new satellite platforms such as Sentinel 2 (Drusch et al., 2012) and even CubeSats (McCabe et al., 2017b), provide a much higher orbital repeat cycle. Regardless, a satellite driven framework would provide a singular opportunity for improving agricultural water management and monitoring (Brocca et al., 2018; Jalilvand et al., 2019).

Quantifying crop water use via evaporation is a fundamental step towards estimating agricultural water use. In the absence of within-field flow metering or a surface flux monitoring system (Baldocchi et al., 2001), a remote sensing-based approach to estimate land surface evaporation provides a suitable alternative. A comprehensive body of research has been dedicated to developing and intercomparing techniques (Kalma et al. 2008; Fisher et al., 2017) and exploring the application of these from local (Allen et al., 2007; Anderson et al. 2011) to regional and global scales (Miralles et al. 2016; McCabe et al.

2016). Most of these models combine available meteorological data with satellite-based vegetation retrievals, or with vegetation and thermal infrared measurements, to estimate the surface evaporation. For example, Song et al. (2016) and Li et al. (2017) used the two-source energy balance (TSEB; Norman et al. 1995) and the surface energy balance models (SEBS; Su, 2002) respectively, to map evaporation from semi-arid irrigated sites. Aragon et al. (2018) applied the Priestley-Taylor JPL (PT-JPL; Fisher et al., 2008) with ultra-high resolution vegetation data from CubeSats to map evaporation over irrigated fields.

Anderson et al. (2012) demonstrated and discussed the ability of Landsat thermal imagery (e.g. with TSEB) to monitor evaporation and its application for water resources management. TSEB has also been used as part of the Atmosphere-Land Exchange Inverse (ALEXI; Anderson et al., 1997) and its associated disaggregation scheme (DisALEXI; Norman et al., 2003) to generate high-resolution maps of agricultural water use. As it currently stands, there remains no single retrieval technique

that has been identified as the best performing evaporation model across all biomes and scales (Ershadi et al., 2014; Michel et al., 2016; McCabe et al, 2016), with model selection ultimately based on past performance and expert knowledge.

While there has been sustained efforts towards estimating agricultural water use through evaporation modeling, there have been relatively few studies aimed at retrieving actual irrigation amounts for monitoring purposes. For example, Folhes et al. (2009) combined known irrigation values from 40 selected fields with satellite-based evaporation estimates in order to derive an irrigation efficiency. They then used this information to estimate the total water use in a semi-arid irrigated agricultural region in Brazil. Santos et al. (2008) integrated satellite-based evaporation estimates into a water balance model in order to provide improved irrigation guidelines to reduce water use. Another approach that has been explored is to use satellite-based evaporation estimates to constrain the irrigation input into a land surface model (LSM) by way of inverse modeling. Droogers et al. (2010) used synthetic model runs to determine whether this approach could be employed to retrieve irrigation amounts. Importantly, they explored the effects of satellite retrieval frequency on the estimated irrigation, demonstrating that while RMSE increased with larger observation intervals, Landsat data could potentially be used for this purpose. Huang et al. (2015) used a similar approach by assimilating temporal variations of MODIS-derived Leaf Area Index (LAI) and evaporation data into the soil water atmosphere plant (SWAP) model to derive irrigation depth. They found that assimilating both variables resulted in the least relative error of the resulting crop yields compared to official county statistics. Lopez (2018) also used evaporation estimates and an LSM in an inverse modeling approach to retrieve irrigation rates from 40 center-pivot fields, demonstrating the potential for obtaining seasonal irrigation rates for individual fields. More recently, Jalilvand et al. (2019) explored the possibility of using an approach designed to retrieve rainfall from satellite-based soil moisture data in order to infer irrigation amounts, expanding upon the work of Brocca et al. (2018). However, this approach is limited by the scale at which soil moisture can currently be retrieved (e.g. $0.25°$), as well as by the uncertainty of the techniques used to obtain the actual rainfall amounts that need to be removed (Jalilvand et al., 2019).

To date, the potential of coupling a land surface model with satellite estimates (via evaporation) has yet to be fully exploited for operational field-scale irrigation monitoring. While estimates of evaporation represent the main loss of water from agricultural systems (through soil evaporation and vegetation transpiration), losses through deep drainage are not generally accounted for in evaporation models. LSMs can account for such hydrology, simulating the exchanges of water and heat between the land surface and the atmosphere and providing a detailed water balance that is beyond standard evaporation process models. One simple method to incorporate irrigation in an LSM is to directly add the irrigation rate to the rainfall component (e.g. Ozdogan et al., 2010), but this requires the values of the irrigation rates to be known (or assumed) a priori. However, satellite-based evaporation estimates could be used to constrain the irrigation value needed to reproduce the observed water use. In this study, the idea of combining satellite-based evaporation estimates with LSM simulations to indirectly obtain irrigation rates, and consequently groundwater extraction amounts, was applied over two distinct irrigated agricultural regions. As in the soil moisture-based approach of Brocca et al. (2018) and Jalilvand et al. (2019), rainfall is not explicitly considered in this water balance. The application of this inverse modeling approach can be simplified in arid environments, as irrigation rates are typically not modified when short-duration sporadic rainfall events occur, since the amounts that can be captured by

the crops is limited. Indeed, over the growth cycle of a typical crop, the applied irrigation volume is at least an order of magnitude greater than any rainfall component (Kenawy and McCabe, 2016).

The present study details the first large-scale implementation of this framework, focusing on quantifying groundwater abstractions for a single year to illustrate the feasibility of larger-scale and longer-term implementation. The integrated satellite data and modeling approach is designed to map the extent and distribution of fields, estimate crop water use, and infer groundwater abstraction rates. To do this, the framework exploits an object-based image classification technique (Johansen et al. 2010) for mapping individual fields, where one field is defined as the area covered by a single center-pivot rotating arm. The latter is an important aspect that was required in order to apply the data-modeling framework in parallel, i.e. to effectively obtain irrigation rates over a large region containing thousands of fields. Naturally, this allows the display and aggregation of groundwater abstraction rates, and other relevant information over arbitrary delineations, such as management zones or farms, and represents a novel aspect of this work. To date, there have been no comparable efforts attempting the retrieval of individual irrigation rates for a large number (thousands) of fields, and thus this study represents an effort that moves closer towards big-data driven operational monitoring of individual fields. Our framework provides a novel water accounting system for agricultural management in Saudi Arabia, and offers an independent benchmark for water loss over a region that is routinely omitted in global (and even regional-scale) evaporation products (Mu et al., 2011; Miralles et al., 2011). Covering one of the largest agricultural regions in Saudi Arabia, the study provides a benchmark against which the impact of water policy changes can be evaluated in the future, and demonstrates the potential of broader-scale application elsewhere.

## 2 Description of study regions

The study focused on two different agricultural areas in Saudi Arabia, which enabled an evaluation of the proposed groundwater abstraction estimation framework (Section 3) and subsequent larger-scale application to be explored. To evaluate the performance at the individual field-scale, the strategy was first applied to 40 center-pivots located on a small farm (Section 2.1) southeast of Riyadh. For this site, available irrigation data for the year 2015 was obtained directly from the farm management (Section 3.5) based on in-house field reporting. To assess the large-scale application of our groundwater abstraction estimation strategy, it was then applied to thousands of center-pivot fields in Al Jawf (Section 2.2), one of the largest agricultural regions in Saudi Arabia. With no individual field data available for comparison on this region, the total groundwater abstraction estimates were compared with some previous regional-scale estimates.

### 2.1 The Tawdeehiya experimental farm

The Tawdeehiya farm (Fig. 1) is a medium-sized commercial agricultural facility that consists of approximately 40 center-pivot fields, each with an extent of approximately 50 ha, n.b. within the average field size of those found in the larger Al Jawf region (see Fig. 1 and Section 4.2). The farm is located about 200 km southeast of Riyadh, and exhibits similar environmental and climatic conditions as the Al Jawf study area (i.e. low rainfall and high daytime summer temperatures

exceeding 40°C). Crops grown in this farm during 2015 included a range of vegetables, alfalfa, Rhodes grass and maize, with a total area under cultivation of more than 2000 ha. While one Landsat tile (path/row 165/43) is enough to observe the entire farm, the adjacent tile (164/43) offers additional coverage of fields on the eastern side of the farm. Additional details of the site and data can be found in Lopez (2018), as well as some related remote sensing-based studies that provide further
description (Aragon et al., 2018; Houborg and McCabe, 2018b).

## 2.2 The Al Jawf agricultural region

The Al Jawf province is located in the north of Saudi Arabia (29°–32° N, 36°–41° E) and is one of the top five agricultural regions both in terms of agricultural area and water use (FAO, 2013; Chowdhury et al., 2016). Most of the agricultural area is managed by large commercial farms, the majority employing center-pivot irrigation (Fig. 1). The irrigated
area in Al Jawf has increased significantly over the last three decades, from being practically non-existent in the 1980's, to covering more than 1,500 km$^2$ by 2005 (FAO, 2013), with more than 90% estimated to be irrigated using groundwater delivered via center-pivot systems (Al-Rhumikhani and Din, 2004). The Saudi statistical yearbook (SSYB, 2010) reported a decline in acreage in Al Jawf from 1,600 km$^2$ in 2007 to 1,200 km$^2$ in 2009. However, newer versions of this report (SSYB, 2013) do not offer a regional disaggregation of agricultural area. Under the Ninth Development Plan from the Ministry of Economy and
Planning (MEP, 2010), it was reported that there was a 2.5% annual decrease in agricultural water use in the Kingdom, from 17.5 BCM in 2004 to 15.4 BCM in 2009, which was attributed to regulations to rationalise water consumption and cultivation of water-intensive crops (MEP, 2010). In Al Jawf, the same report forecasts a continued decline in groundwater abstraction from 1.5 BCM in 2009 to 1.2 BCM in 2014. Although other sources include more recent estimates for the total agricultural water demand in the Kingdom (MEWA, 2019), these are not available on a regional basis. The majority of crops grown in Al
Jawf in 2009 (SSYB, 2010) were cereals (60% in area; with wheat being the most predominant), followed by fruits (24%, consisting of dates, grapes and citrus), fodder (11%; mostly clover) and vegetables (4%; mostly potatoes and tomatoes). Unfortunately, the same level of detail is not available from more recent reports (SSYB, 2013), but we note that cereal production is no longer supported to any significant extent. One of the challenges for attributing water use to crop-type in this region is the lack of available ground-based land cover data. Al-Rhumikhani and Din (2004) used a data set from one
agricultural site in 2001 that cultivated alfalfa, potatoes, tomato, and wheat, to classify crops in Al Jawf from Landsat imagery. To our knowledge, no other recent crop classification efforts have been made in the region.

Although rain-fed agriculture represents about 10% of the cultivated area within Saudi Arabia, this is limited to the south-west regions of Jizan, Baha, Aseer and Makkah (FAO, 2008a). Annual rainfall values in Al Jawf are, as in most of Saudi Arabia, less than 50 mm/year (Kenawy et al., 2016), and consequently there is insufficient water to support rain-fed irrigation.
Indeed, agriculture in Al Jawf is entirely supported by groundwater extraction (Al-Rhumikhani and Din, 2004; MEP, 2010; Chowdhury et al., 2016). In 2015, average wind speeds in this region were relatively low (less than 5 m/s) throughout the year, but reached maximum speeds up to 16 m/s (the meteorological data used in this study is described in Section 3.4.3). Average temperatures ranged from about 10 °C in January and December, up to 32 °C in July and August, and were consistently higher

than 25 °C from May to October. Maximum temperatures of up to 44 °C occurred in August, while the minimum temperature reached was -1 °C and occurred in January. Relative humidity ranged from about 25% from May to September, and around 55% during January, November and December.

### 3 Pivot-based groundwater abstraction framework

5        A key output of this study is the estimation of groundwater abstraction from thousands of individually delineated center-pivot fields. At the core of the methodology is the indirect estimation of the volume of irrigation that needs to be applied to a field in order to reproduce the satellite observed crop water use. Figure 2 presents a schematic of the methodology with the necessary inputs, intermediate processes and relevant outputs. In this approach, the first step is an automated processing framework (Section 3.1) that performs Landsat image acquisition, cloud and cloud shadow detection, regionally optimised

atmospheric correction, and finally higher-level product generation of albedo, Normalized Difference Vegetation Index (NDVI), leaf area index (LAI) and land surface temperature (LST). This procedure was based on recent efforts in combining machine learning techniques with physically based model inversion results (Houborg and McCabe, 2018a), which have made the automatic estimation of these parameters over large arid regions possible. The data obtained in this crucial first step was then directed into both the evaporation and land surface models (see Section 3.4). The next step uses a geographical object-

based image analysis (GEOBIA) procedure (Johansen et al., 2010) to detect and map individual center-pivot fields based on NDVI data (Section 3.2). Meteorological data were retrieved from a reanalysis performed using the Advanced Research WRF (ARW; Skamarock et al., 2008) model over the Arabian Peninsula. Details of this dataset are described in Langodan et al. (2014); Viswanadhapalli et al. (2017, 2019); and Dasari et al. (2019), so only a brief description is provided in Section 3.4.3. To provide needed estimates of crop water use (via evaporation), the TSEB model (see Section 3.4.1) was run over two Landsat

tiles on each study region (172/39 and 171/39 for al Jawf; 165/43 and 164/43 for the Tawdeehiya Farm) using the WRF data together with the Landsat-based vegetation and biophysical parameters (at 30 m spatial resolution and at a 16 day frequency) obtained in the first step.

        Up to this point, all processes involved use of the entire Landsat tiles. The following steps in the framework were performed independently over each separately delineated center-pivot field (herein simply referred to as "field"). These

operations were performed in parallel using hundreds of CPUs on a high-performance computing cluster (see https://www.hpc.kaust.edu.sa/ibex for further details). To do this, data were first extracted for each field using Geospatial Data Abstraction Library (GDAL) tools. Next, LAI spatio-temporal information was used to detect the seasonal activity of each field. This included the possibility that one field actually contained two different crops with different growing periods, a practice that was recognised by analysing the images, as well as on-site observations. Details of the automated detection

capability developed within this framework are provided in Section 3.3. After this analysis, a "field temporal use index" was computed, defined here as the percentage (in days/days), that the field was used to grow a crop within the year of study. This index, referred to as field use (%), is required to compare irrigation practices among different farms and/or regions.

The Community Atmosphere Biosphere Land Exchange model (CABLE; Kowalczyk et al., 2006; Section 3.4.2) was used as an offline land surface hydrology model to indirectly estimate the irrigation rate applied over each season, for any particular field, for the study year of 2015. As rainfall is limited in this region, implementing CABLE simulations without an irrigation component leads to an almost complete lack of evaporation signal. TSEB information on evaporation was used to

infer the irrigation amount that would be needed in CABLE to reproduce the estimated evaporation. In this sense, TSEB improves the estimation of CABLE by providing the missing irrigation component in this region. The procedure is done as follows: for each active season identified in a field, an ensemble of CABLE runs were performed under different irrigation amounts. Denoting $E_{sat}$ as the remotely sensed estimates of evaporation (obtained using TSEB; see Section 3.4.1) and $E_{LSM}$ as the output evaporation from the CABLE land surface model, a cost function that was proportional to the difference between

$E_{LSM}$ and $E_{sat}$ was accumulated during each available observation of $E_{sat}$:

$$J = \sum_i [Y(t_i) - H(X(t_i))]^T [Y(t_i) - H(X(t_i))], \tag{1}$$

where $H$ is an operator that maps the land surface model space $X(t_i)$ to the observation space $Y(t_i)$ n.b. the satellite estimates need not be available at the same resolution as the model, or might be incomplete (e.g. due to the presence of clouds). Lopez (2018) used a stochastic optimization approach (Spall, 1998) that iteratively updates the irrigation rates to minimise the

objective function. That method required tuning two parameters that control the speed at which the update occurs, a process that was initially performed by trial and error. Importantly, the number of fields that were evaluated in Lopez (2018) was significantly smaller (40) than the present study (more than 5000). Unfortunately, the transferability of the optimization parameters to a larger number of fields was limited, as the optimised irrigation rates either diverged, or the values did not update at all due to improper scaling of the gradient of the cost function. Under the rationale that the search space is relatively

small, i.e. by constraining the irrigation rate to realistic values (e.g. 1.1 – 3 times the observed evaporation rates), a simple exhaustive search (brute-force) was implemented in this study. This removed the need for a trial and error approach for optimization of parameters, as well as the need to compute a gradient (which requires at least two model runs for each step), i.e. trading off precision for improvements in computation. The latter was a key consideration to efficiently apply this methodology to thousands of fields.

At the end of the process, the irrigation rate that produced the most accurate evaporation estimate (compared to TSEB estimates) was the one used to calculate the total groundwater abstraction over the field. The irrigation rate (Irr) applied to CABLE is the actual amount of water reaching the ground, and it was assumed to be a constant fraction (1-$C_{loss}$) of groundwater abstraction ($G_w$) as in Eq. (2):

$$Irr = G_w (1 - C_{loss}). \tag{2}$$

The loss term was calculated using the following rationale. The amount of water pumped out of the well ($G_w$) is sprayed by nozzles positioned on a rotating arm irrigating the field. A fraction of this water ($C_{loss}$) is lost due to wind carrying moisture out of the field to the desert soil or to other fields. Under similar conditions (hyper-arid irrigated fields), this fraction has been estimated to be between 12 and 20% (Steiner et al., 1983; Sadeghi et al., 2017). Without frequent on-site measurement of this

loss value across the whole field, it is not possible to incorporate it directly into the model. In this study, the aim was to provide an estimate of the irrigation amount (Irr) as an approximation of the groundwater abstraction ($G_w$). Efforts to translate this into actual groundwater abstraction on a regular basis for large areas are necessary, but will require ground observations or further model improvement. For this study, a conservative value (in time and throughout the fields) of 20% ($C_{loss}= 0.2$), and thus a

factor of 1.25 i.e. $(1-C_{loss})^{-1}$, was used to scale irrigation to abstracted groundwater.

In terms of the overall field water balance, the approach employed in this study was simplified by the fact that rainfall in this region does not represent a significant source that can be used by crops. As such, the rainfall component in CABLE was replaced by the irrigation rate that is being estimated in the iterative procedure. The validity of this assumption and thus the applicability of this model is certainly reasonable in most of the major agricultural regions of Saudi Arabia, with the exception

of regions with significant rain-fed agriculture, such as those in Jizan and Baha. However, even in those regions, the annual rainfall rarely exceeds 300 mm, and occur within a relatively well-defined period of the year. For example, coffee production is an important economic activity in Jizan, occurring mainly in high altitudes within the Fifa mountains, and farmers there regularly require groundwater as an additional source to meet the water needs of coffee trees, especially during extended periods of drought (Al-Abdulkader et al., 2018; Sayed et al., 2019). In other regions, the contribution of rainfall might be

proportionally higher and thus need to be removed before obtaining the irrigated amount.

### 3.1 Vegetation indices and biophysical parameters retrieved from satellite data

NDVI is a widely used metric describing surface greenness (Tucker, 1979; Beck et al., 2011) and is computed herein directly from the Landsat near infrared (851-879 nm) and red bands (636-673 nm). Prior to computing NDVI, Landsat images for the year 2015 (between 20 and 22 scenes were acquired for each of the tiles used in this study) were atmospherically

corrected to surface reflectances using a regionally optimised Second Simulation of the Satellite Signal in the Solar Spectrum (6S; Kotchenova et al., 2006)-based approach (Houborg and McCabe, 2017). Cloud and cloud shadow detection was performed using the Function of mask (Fmask) algorithm (Zhu and WoodCock, 2012). Another biophysical indicator for vegetation growth monitoring is the LAI, defined as the projected area of leaves over a unit of land area. LAI is a key parameter that has been used to improve water and energy flux modeling over agricultural fields (Aragon et al., 2018). However, as opposed to

NDVI, LAI cannot be computed directly from satellite data. While simple relationships between LAI and other vegetation indices (including NDVI) have been used (Turner et al., 1999; Colombo et al., 2003; Fan et al., 2009), the applicability of such relationships for regions other than where they were developed (using ground measurements) has been brought into question (Wang et al., 2005; Atzberger et al., 2015; Kang et al., 2016). Houborg and McCabe (2018a) used a machine learning approach to develop relationships between LAI and several vegetation indices over a desert agriculture site in Saudi Arabia (Tawdeehiya

farm; Section 2.1). They used a combination of in situ measurements and physically based model inversion results as a hybrid training dataset, with retrievals showing good performance compared to the in situ LAI measurements. In our study, a coupled leaf-canopy model (PROSAIL) produced forward runs over a wide range of realizations, and these were used as a training dataset to develop estimates of LAI using a Random Forests (RF) approach. The inversion of the forward runs needed to derive

LAI was based on the REGularized canopy reflectance model (REGFLEC; Houborg et al., 2015) which has been shown to be suitable for largely automated applications (Houborg and McCabe, 2016). The configuration of REGFLEC in this study followed Houborg and McCabe (2018a), but using only the model inversion results (e.g. PROSAIL) due to a lack of in situ LAI data in the larger region of Al Jawf. PROSAIL combines a leaf optical properties model (PROSPECT; Jacquemoud and Baret, 1990) with a canopy bidirectional reflectance model (SAIL; Verhoef, 1984), and has been used to retrieve LAI for a wide range of crops (Jacquemoud et al., 2009; Vohland et al., 2010; Rivera et al., 2013). The RF ensemble-based decision tree technique was used to learn the complex non-linear associations between the spectral data and the target biophysical property (i.e. LAI). In this study, the 'ranger' RF package in R was used for model training and prediction. This package is optimised for efficient memory usage with large and high dimensional datasets: a critical aspect for working with Landsat tile domains. Figure 3 shows the resulting NDVI and LAI maps for one Landsat tile (path/row 172/39) on April 23, 2015, demonstrating the high contrast between the bare desert soil and irrigated agriculture and the within-field variability that can be observed at these high resolutions. Most of the agricultural fields in Al Jawf (95%) were located within this Landsat domain, however the adjoining path/row 171/39 was also included to give a complete account of agriculture in Al Jawf (see Fig. 1). The Tawdeehiya Farm is located on the eastern edge of path/row 165/43, and hence some fields can also be observed by path/row 164/43.

## 3.2 Semi-automatic delineation of center-pivot irrigation fields using Landsat imagery

A GEOBIA approach was developed in the eCognition Developer 9.3 software to delineate individual fields for the study sites. As seasonal crop cycles prevent all active fields within a specific year from being detected at a single point in time, the full Landsat image time series was used for 2015. Two layers based on the annual image time series (20 - 22 images) were produced: (1) a maximum NDVI layer and (2) a minimum panchromatic layer. NDVI was first calculated for all images in the time series, and the maximum NDVI value for each pixel in the time series was assigned to the final maximum NDVI layer. Similarly, all the panchromatic images within the time series were used to produce a single panchromatic layer, representing the minimum value within the time series. To detect all active fields in 2015, a multi-threshold segmentation was applied to cluster all pixels with a maximum NDVI value of > 0.20 together and classify these objects as "vegetation". Unclassified pixels surrounded by "vegetation" objects were first merged with the respective "vegetation" objects, and these objects were then classified as center pivots, if the object length was ≤ 1,200 m, length/width ratio ≤ 1.1, and elliptic fit ≥ 0.90. The length/width ratio and elliptic fit features were used to identify round objects, while the length feature ensured that neighboring fields merged together were not initially classified as fields before they had been separated into objects, representing an individual field.

The minimum panchromatic layer was subjected to an edge extraction Lee Sigma filter. This filtering process produced another layer, highlighting bright edges in the imagery, i.e. areas with large contrast in panchromatic pixel values, such as the edges between center pivots and surrounding sandy soil. Separating adjoining fields required several processing steps to first identify pixels with high edge filtering values and the use of region-growing algorithms to grow these high value edge filtered pixels into neighboring pixels with lower values. This allowed most of the adjoined fields to be separated.

Some refining of the delineation results were then performed to ensure the fields were extended to their perimeter, which was done using a number of object shape criteria and an NDVI threshold of 0.20. While most of the fields (approximately 85%) were correctly classified at this stage, there were still several half or "Pac-man" shaped fields remaining to be classified. These remaining fields were manually delineated, followed by an object growing and object shrinking process to refine the manually delineated field edges.

## 3.3 Irrigation activity detection using LAI

LAI time series were used as an indication of vegetation growth to estimate periods of active irrigation, which were then used to constrain the start and end dates of CABLE model runs. However, upon visual analysis of multiple fields within the study regions, we observed that fields are often divided into two halves, each with its own crop and potentially different irrigation amounts. Therefore, prior to obtaining a representative LAI time series for analysis, it was necessary to further delineate the two sections of the field if it was indeed divided. To achieve this, the k-means clustering algorithm was employed. In general, the idea of clustering is to identify groups of objects that are similar to one another and different from those in other groups (Jain and Dubes, 1988; Jain et al., 1999). The k-means algorithm (MacQueen, 1967) is a partitioning clustering method (i.e. there is no overlap between the groups) that has been widely used in remote sensing studies. For this purpose, let $X = \{x_i\}, i = 1,2,3, \dots, n$ be a set of n-dimensional points to be clustered into a set of K clusters $C = \{c_k\}, k = 1,2,3, \dots, K$ (in our study, K is set to 2). The squared error between the mean of each cluster and the points within the cluster can be computed by Eq. (3):

$$J(c_k) = \sum_{x_i \in c_k} ||x_i - u_k||^2. \tag{3}$$

The aim of the K-means algorithm is to minimise the sum of $J(c_k)$ for k = 1,2,3,…, K. Four main steps are followed iteratively to minimise this sum: (1) select K points as the initial centroids, (2) form K clusters by assigning all points to the closest centroid, (3) re-compute the centroid of each cluster, and (4) repeat (2) and (3) until there is no significant change in the centroids within consecutive iterations. To speed up convergence, the "k-means++" (Arthur and Vassilvitskii, 2007) algorithm was used to select the initial cluster centers. A more detailed description of the k-means algorithm is provided by Jain (2010).

Prior to applying the clustering algorithm, it was more efficient to extract the features to represent the characteristics of the time series. To do this, the discrete wavelet transform (DWT) was employed, which is an efficient procedure used to separate deterministic from stochastic components of a signal (Heil and Walnut, 1989). The DWT has been used to analyse satellite images in the context of noise reduction as well as change detection (Zhu and Yang, 1998; Wang and Paliwal, 2006; Martínez and Gilabert, 2009). The main idea behind application of the DWT is that the signal is represented as a combination of approximation and detail coefficients (Heil and Walnut, 1989):

$$x(t) = A_J(t) + \sum_{j=1}^{J} D_J(t) , A_J(t) = \sum_{k=1}^{n2^{-J}} a_{Jk}\phi_{Jk}(t) , D_j(t) = \sum_{k=1}^{n2^{-j}} d_{jk}\psi_{jk}(t), \tag{4}$$

where AJ(t) and Dj(t) are respectively the approximation and detail coefficients, and J is the decomposition level. The detail coefficients are generated by projecting the original signal x(t) using a set of wavelet basis functions defined as $\psi_{j,k}(t) =$

$\sqrt{2^{-j}}\psi(2^{-j}t-k), j=1,\ldots,J, k \in Z$, where k is the shift parameter and is the base function. In other words, the detailed signal Dj(t) at level j is generated by the detailed signal Dj(t) at scale j and can be obtained by applying a high-pass filter (g) on the original and scaled signals. In a similar way, the approximation coefficients are generated by projecting the signal on a set of orthonormal scaling functions given by $\phi_{j,k}(t) = \sqrt{2^{-j}}\phi(2^{-j}t-k), j=1,\ldots,J, k \in Z$. Similarly, the scale signals are

computed by applying a low-pass filter (h) on the original and scaled signals. Gao and Yan (2010) provide a more detailed description of the DWT.

       In our study, LAI time series were first transformed into DWT components by level-1 decomposition of the basis function 'haar'. Then, to establish whether the field is divided into two parts or not, two threshold values were used. Both values relate to the *cosine similarity* (Eq. 5), which measures the similarity among pixels of the same field:

$similarity = \dfrac{A \cdot B}{||A|| \, ||B||} = \dfrac{\sum_{i=1}^{n} A_i B_i}{\sqrt{\sum_{i=1}^{n} A_i^2} \sqrt{\sum_{i=1}^{n} B_i^2}}$,                                                           (5)

where $A_i$ and $B_i$ are components of a vector A and B respectively (i.e. one vector represents the time series of one pixel). The cosine similarity values of two vectors ranges from -1 to 1. The closer the value approaches 1, the closer the direction of the two vectors. On the other hand, the closer the value gets to -1, the more the two vectors go in the opposite direction. For each pair of pixels, the cosine similarity of DWT components of time series were first computed, and then the time series of the two

pixels were determined to be similar or not by using a threshold of the cosine similarity (tcs). The second threshold used was based on the fraction of pairs where the value of cosine similarity was higher than tcs, i.e. defining nTcs as the number of pair values with similarity higher than tcs, and N as the total number of pairs, the ratio to use as the second threshold was tpcs = nTcs/N. Upon an exploratory analysis based on a representative sample of a few hundred fields, the appropriate values for these two thresholds were determined as tcs = 0.75 and tpcs = 0.8. Hence, a field was classified as partitioned when the tpcs

value was exceeded, and was thus further processed by the k-means algorithm for clustering.

       Next, a number of representative pixels within the field (or sub-field if partitioned) were selected and used to determine the start and end dates for each season within the land surface hydrology model runs. The pixels were selected based on a criteria that the daily LAI values, obtained by linearly interpolating from the 16-day LAI time series, were consistently within the interquartile range (e.g. 25% and 75%). This was done to remove the influence of outlier pixels. Finally, the mean

value over these pixels each day was taken as the mean LAI time series (*mLAI*). Using this mean LAI time series, crop growing seasons were then selected based on the start and end dates of the first season, defined by the period where *mLAI* is higher than a threshold *tLAI* (in $m^2/m^2$). A value of *tLAI* = 0.3 $m^2/m^2$ provided a reasonable delineation of growing seasons for a large number of fields. Furthermore, to remove noise that could result from the interpolation of the LAI time series, only seasons that were at least 30 days in duration were processed, which is the approximate length of the shortest crop growing seasons

observed for alfalfa crops and also ensures that it includes at least one Landsat scene.

       Upon analysis of the shape of *mLAI*, in terms of the number of peaks as a measure of the oscillatory nature of the time series, fields were then classified into two possible categories: seasonal or perennial. A "seasonal" field had clearly defined

growing cycles that were separated by inactive periods. Figure 4 (a) shows a field that was active for only three months, with clear start and end dates obtained by computing the dates at which the mean seasonal pixels in the LAI time series intersected with 0.3 m$^2$/m$^2$. Perennial fields had vegetation patterns that reflected long periods (up to year-round) of vegetation above the LAI threshold, but with intermittent cut and re-growth periods. This is typical of a field growing alfalfa or grass, where the

production is continuous throughout most of the year. An example of a perennial field is shown in Figure 4 (b). This field was active from April to December 2015, but with intermittent cut/re-growth cycles. For this second category of crops, it was not straightforward to select clear periods for retrieving irrigation, due to Landsat's temporal resolution of 16 days. To ensure that there would be enough satellite evaporation estimates for constraining the land surface model runs, the process for these types of pivots was performed on a quarterly basis.

**3.4 Models and ancillary data**

**3.4.1 Satellite evaporation estimation using TSEB**

The TSEB model (Norman et al., 1995) was used to obtain the satellite-based estimates of evaporation that constrain the land surface hydrology model runs. TSEB was selected based on its proven utility in the estimation of evaporation over irrigated crops in semi-arid and arid regions (e.g. Colaizzi et al., 2012; Zhuang and Wu, 2015; Nieto et al., 2019). TSEB is

based on the energy balance ($R_n = LE + H + G$), where R$_n$ is the net radiation reaching the crop canopy and soil surface, H is the sensible heat flux (i.e. the energy transformed to heat and released into the atmosphere), G is the soil heat flux, and LE is the latent heat flux of evaporation, which is the key link between the energy and water balance equations. In TSEB, the LE term is obtained as a residual of the energy balance equation, and along with H and R$_n$, is divided into separate components for the soil and canopy at each pixel. The model for sensible heat flux in TSEB is based on a network of temperature gradient-

transport resistances between the air, canopy boundary layer, canopy and soil. In this study, the "in-series" resistance scheme (Shuttleworth, 1985) for the sensible heat flux was used. The in-series scheme was selected as it has been demonstrated to estimate heat fluxes of densely vegetated areas more accurately than the parallel or patch schemes (Kustas et al., 1999; Colaizzi et al., 2012). In center-pivot fields, the crops are not structured in rows, and at maturity the canopy covers the entirety of the soil surface (with the exception of the beam tracks of the pivot), and thus the area was considered as densely vegetated. The

in-series approach accounts for the coupling between soil and canopy heat fluxes.

The sensible heat flux is defined below for canopy (H$_c$) and soil (H$_s$) in Eq. (6):

$$H_c = \rho c_p \left(\frac{T_c - T_{AC}}{r_x}\right) ; H_s = \rho c_p \left(\frac{T_s - T_{AC}}{r_s}\right), \tag{6}$$

where T$_c$ is the canopy temperature, T$_s$ is the soil temperature, T$_{AC}$ is the temperature of the canopy-air space, r$_x$ is the canopy boundary layer resistance, r$_s$ is the resistance of air between the soil surface and source height, $\rho$ is the density of water, and

$c_p$ is the specific heat of water. The calculation of the two resistances r$_s$ and r$_x$ was done following the methods of Sauer et al. (1995) and McNaughton (1995) respectively.

The LE fluxes for the canopy and soil are determined based on initial estimates for $T_c$ and $T_s$, which are then iteratively refined until $LE_s$ is positive (or after a maximum number of iterations). An initial estimate of $LE_c$ (canopy latent heat flux) is required in order to obtain the values of $T_c$ and $T_s$. This is done using the Priestley-Taylor equation and then solving for $T_c$:

$$T_c = T_a - (R_{nc} - LE_c)r_{ah}/(\rho c_p), \tag{7}$$

where $T_a$ is the air temperature from the WRF data and $r_{ah}$ is the aerodynamic resistance to heat transport. The value of $r_{ah}$ depends on atmospheric stability parameters, wind speed, and measurement heights for temperature and wind speed, which is set to WRF's near surface level (2 m in this study). The aerodynamic roughness length for heat and momentum transport were defined as $z_{0M} = H_C/8$ and $z_{0H} = \frac{z_{0M}}{\exp(k_B{}^{-1})}$, where the $k_B{}^{-1}$ (i.e., $\ln(z_{0M}/z_{0H})$ ) parameter was set to 2 as in Norman et al. (1995). Without detailed in situ information describing land cover and crop development stage, canopy height was prescribed to a constant value of 0.3 m: a reasonable assumption based on the typical crops such as alfalfa, wheat, and vegetables being grown in this region.

The net radiation is computed by Eq. (8):

$$R_n = (1 - \alpha)S_{dn} + L_{dn} - \varepsilon_{surf}\sigma_b LST^4, \tag{8}$$

where $\alpha$ is the albedo (computed from Landsat data as in Liang, 2000), $S_{dn}$ and $L_{dn}$ are the incoming shortwave and longwave radiation components (derived from WRF data), $\varepsilon_{surf} = (f_\varphi \varepsilon_{veg} + (1 - f_\varphi)\varepsilon_{grd})$ is the surface emissivity with $f_\varphi$ described in Equation 11 and $\varepsilon_{veg} = 0.98$ and $\varepsilon_{grd} = 0.93$ are the canopy and soil emissivities respectively, and $\sigma_b$ is the Stefan–Boltzmann constant.

The net radiation that reaches the canopy $R_{nc}$ is modeled as:

$$R_{nc} = R_n \left(1 - e^{-0.45 LAI/\sqrt{2\cos(\varphi_z)}}\right), \tag{9}$$

where $\varphi_z$ is the solar zenith angle. This simple parameterization for $R_{nc}$ was developed based on the Cupid model for dense canopies as described in Zhuang and Wu (2015). The initial soil temperature is then given by Eq. (10) and is updated iteratively as described in Norman et al. (1995):

$$T_s \cong \sqrt[4]{(T_r^4 - f_\varphi T_c^4)/(1 - f_\varphi)}. \tag{10}$$

In this study, the radiometric temperature ($T_r$) is the LST derived from Landsat's thermal infrared sensor data, with atmospheric correction of the at-sensor brightness surface temperature performed using MODTRAN (Berk et al., 2005; Rosas et al., 2017). In this process, atmospheric profile data from MERRA2 (Gelaro et al., 2017) was used and emissivity fields were based on the methods of French et al. (2005) using estimates of vegetation fraction (Anderson et al., 2007). A Data Mining Sharpener (DMS) technique based on regression tree analysis (Gao et al., 2012) was used to perform spatial sharpening of LST to 30 m resolution. The vegetation fraction at the sensor view angle $\varphi$ (0 for Landsat) is:

$$f_\varphi = 1 - exp\left(\frac{-0.5\ LAI}{\cos(\varphi)}\right). \tag{11}$$

The vegetation fraction assumes a spherical leaf angle distribution, which is a good approximation for general plant canopies, spreading the leaf area uniformly across solar zenith angles (Campbell 1998). For Saudi Arabia, at the Landsat overpass time, the extension coefficient Kb is approximately equal to 0.5.

For pixels with low LAI values (LAI < 1) i.e. where the soil component is dominant, the canopy component was omitted by applying a simpler, one-source energy balance (OSEB). In the OSEB, the sensible heat flux is estimated by using a one-layer circuit network. Although this can lead to a sharp transition in ET for values around LAI=1, this was done in order to reduce the influence of high bare soil evaporation values using TSEB, which would cause an overestimation of water use in fallow or inactive fields and where the surface can be modelled as a single layer. In OSEB, H is simply given by:

$$H = \rho c_p \left( \frac{T_r - T_a}{r_{ah}} \right). \tag{12}$$

The soil heat flux (G) model of Santanello and Friedl (2003) was used in this study. This model includes a simple relation describing the covariation between daytime ground heat flux and net radiation ($R_n$):

$$G = A cos \left( \frac{2\pi(t + 10{,}800)}{B} \right) R_{ns}, \tag{13}$$

where A represents the maximum ratio of $G/R_{ns}$, B is a constant that minimises the divergence of the equation to that of measured values, and t is the number of seconds between the satellite overpass time and solar noon. The values of A and B were left as in the original parameterization, with A = 0.31 and B = 74000 s.

The use of TSEB to estimate evaporation has been validated in similar arid regions (Colaizzi et al., 2012; Zhuang and Wu, 2015; Nieto et al., 2019). With a lack of sufficient ground-based measurements in our study regions, we did not attempt to provide a new validation data set for TSEB. However, we used data from one eddy-covariance tower installed in 2016 on a center-pivot irrigated field in Tawdeehiya farm (Figure 1; Section 2.1), which showed good estimation for operational purposes in this data-limited region (Figure S1).

### 3.4.2 CABLE

CABLE was used as a stand-alone (offline) land surface hydrology model to estimate irrigation rates based on the evaporation estimates obtained with TSEB. CABLE was selected given its application in other dryland environments and as the land surface scheme for regional and global climate models (Zhang et al., 2009; Haverd et al., 2013; Hirsch et al., 2019), but the approach described in this study is not limited to any particular scheme. Energy and water interactions are modeled in CABLE across six layers of soil (with thickness from the surface to bottom layer of 2.2 cm, 5.8 cm, 15.4 cm, 40.9 cm, 108.5 cm, and 287.2 cm), the canopy layer, and the atmosphere. Similar to TSEB, the heat fluxes in CABLE are modeled by a network of aerodynamic resistances. The sensible and latent heat fluxes are also partitioned into a flux from the soil to the canopy, and from the canopy to the atmosphere. However, in CABLE, the canopy layer is further divided (two-leaf canopy model) between sunlit and shaded leaves, using an approach developed by Leuning et al. (1995).

Another feature of CABLE is that the canopy transpiration, as well as root water extraction, both depend on whether the canopy is wet, dry, or partially wet (Lai and Katul, 2000). A coupling between the root water extraction and stomatal

conductance was added by Haverd et al. (2016), enabling a "root shut-down" that tests for over-extraction in each soil layer, which would otherwise result in high water use efficiencies under drying conditions. Hydraulic redistribution (Ryel et al., 2002) was also added to CABLE in order to improve the representation of the water flux between soil layers. This component involves redistribution of water by roots, and depends on the root density and the rhizosphere conductivity. This term was added as an additional term in Richard's equation, which describes the soil moisture ($\theta$) flux and is based on the one-dimensional conservation equation and Darcy's Law:

$$\frac{\partial \theta}{\partial t} = -\frac{\partial}{\partial z}\left(K + D\frac{\partial \theta}{\partial z}\right) + F_w(z),$$ (14)

where K is the hydraulic conductivity, D is diffusivity, and $F_w(z)$ includes water lost due to soil evaporation, root extraction, or water gained or lost in a layer by hydraulic redistribution (Ryel et al., 2002). In this study, CABLE version 2.3.4 was used. This version of CABLE is freely available at *trac.nci.org.au/trac/cable* after registration. Detailed descriptions of the model are available in Kowalzcyk et al. (2006), Wang et al. (2011), Kowalzcyk (2013), and Haverd et al. (2016).

To support the generality of our approach, the offline global simulations in this application of CABLE use look-up tables for soil and vegetation classification. By default, CABLE includes a soil classification table derived from Zobler (1999), and vegetation types defined by the International Geosphere and Biosphere Program (Loveland et al., 2000). CABLE auxiliary files also include monthly LAI data derived from MODIS data averaged from 2002 to 2009 (Gao et al., 2008; Ganguly et al., 2008), as specified in the CABLE user guide (Srbinovsky et al., 2013). In this study, the default soil texture was used, and assumed as uniform across the study region (the soil corresponds to sandy loam soil in the irrigated regions explored here). However, LAI data in this study was derived as described in Section 3.1, as the coarse-scale MODIS-derived dataset is not representative of the actual crop growing patterns. The possibility of different crops and crop rotation in the same field within the year was considered, as explained in Section 3.3, using the clustering technique based on LAI data. One limitation of the framework is the lack of a crop identification module, which would improve the definition of vegetation characteristics. Here, vegetation parameters were assigned based on the default CABLE cropland vegetation class, as currently no crop identification strategy was implemented, other than the delineation and clustering technique. Finally, CABLE was forced with hourly meteorological data from a WRF reanalysis (Section 3.4.3).

Under basin-scale water budget studies, a spin-up of the model is generally required to achieve a realistic initial soil moisture state. This is normally done by running a representative year of meteorological data several times, or running several years prior to the start of the study period (Ajami et al., 2014; 2015), and assuming that the spin-up period is representative of the "normal" conditions. This assumption does not hold in our simulations because we are aiming to retrieve irrigated amounts, which could change from one year to the other, as different crops are grown. Therefore, this posed a challenge for representing the initial state of irrigated agricultural fields at the start of our simulations. In our study, the spin-up for each field was performed as follows: after estimation of the irrigation amount for one season, the model is run using this irrigation amount, and the final state is saved as the initial state for the next iterative process. However, the problem still lies with the spin-up of

the first period. To solve this, we started by first running the groundwater abstraction strategy for a three-month period prior to the start of the study period, thus generating an initial state for the actual period of study.

### 3.4.3 Meteorological data

The meteorological data required to drive the TSEB and CABLE models were derived from a numerical weather prediction simulation of the Weather Research and Forecasting (WRF) model: specifically, the Advanced Research WRF (ARW; Skamarock et al., 2008) model version 3.7.0. The regional simulation, performed over the entire Arabian Peninsula and neighboring regions, used dynamical downscaling of global analysis data from the National Centers for Environmental Prediction (NCEP). Dynamical downscaling refers to a method to generate a higher spatial and temporal resolution regional climatic model by assimilating available regional datasets, and initialised from coarser reanalysis data (Giorgi and Mearns, 1991; Wilby and Wigley, 1997; Viswanadhapalli et al., 2017; 2019; Dasari et al., 2019). This is typically done by nesting a high spatial resolution domain within a coarser domain. Observational data used in this regional climate model included quality controlled data from the NCEP Atmospheric Data Project (ADP) such as surface station data, wind data from the Quick Scatterometer (QSCAT), WindSat and ASCAT scatterometers, and atmospheric motion vectors from geostationary satellites. The methodology, model parameters, and model physics followed a similar approach as described in Jiang et al. (2009), Langodan et al. (2014, 2016), Viswanadhapalli et al. (2017) and Dasari et al. (2019). The simulations were performed over a 5-year period (2011 – 2015) at an hourly time step, and with the internal model domain having a spatial resolution of 3 km, covering the Arabian Peninsula.

### 3.5 Evaluation of model performance

To evaluate the modeling approach across thousands of individual field sites would require the provision of an extensive data set of on-ground water-use measurements, ideally collected from numerous pivots and over an extensive period of time. However, detailed abstraction, irrigation and crop-water use data in such quantity rarely exists in even the most well monitored sites, let alone for developing country applications. Although we could not collate comprehensive and spatially distributed evaluation data for the Al Jawf study region, we utilise data from the smaller-scale Tawdeehiya farm to provide farm-reported data for the year 2015. The available evaluation data consisted of monthly values of total irrigation application time (in hours), and a single value of flow rate in gallons per minute for each field. To convert to groundwater abstraction (GWA), the flow rate was multiplied by the number of minutes irrigated in each month and converted to units of millions of cubic meters (MCM). The model's performance to quantify field groundwater abstraction was evaluated using the Nash-Sutcliffe efficiency, given by Eq. (15):

$$\text{NSE} = 1 - \frac{\sum_{i=1}^{N}(S_i - O_i)^2}{\sum_{i=1}^{N}(O_i - \bar{O})^2}, \tag{15}$$

where O represents the observations (farm data) and S our estimations, both in MCM for the year 2015. NSE values can be negative (from -∞) or a value from 0 to 1, where 1 represents a perfect match between estimates and observations, and a value of 0 represents that the estimates are as good as the mean of the observed values ($\overline{O}$).

## 4 Results

The strategy as described in this study (Section 3) was applied to two study regions. Section 4.1 presents the results of groundwater abstraction (GWA) of 40 fields at the Tawdeehiya farm and evaluates the performance based on farm data for these fields. Next, Sections 4.2 and 4.3 focus on the larger-scale application of the methodology across the Al Jawf region, demonstrating the framework's capability in terms of information it can reveal at a regional level. First, Section 4.2 explores patterns of irrigation activity e.g. how much time are these fields active throughout the year; whether there is a preference for
seasonal or perennial type of fields (as defined in Section 3.3); and the spatial distribution of yearly groundwater abstraction and field use within the region. Finally, Section 4.3 demonstrates the range of monthly to annual water use in the region, i.e. irrigation rates and derived groundwater abstraction values. A comparison of evaporation estimated by CABLE (with inferred irrigation based on our approach) and TSEB is shown in Figure S2. As a result of this work, a first of its kind spatially distributed map of field-based groundwater abstraction was created as a key output of the monitoring strategy.

**4.1 Pivot-based framework performance at the Tawdeehiya farm**

          In order to evaluate the performance of the approach, the pivot-based water accounting methodology (described in Section 3) was first applied to the 40 active fields at the Tawdeehiya farm (Fig. 1). Seventeen fields were identified as following a perennial planting pattern, with yearly field use values around 86%. Twenty-three fields were classified as seasonal fields, with average field use vales of 57%. The performance evaluation was based on a comparison of the estimated yearly
groundwater abstraction rates and farm-based data reports of flow rates delivered to the irrigation booms. Upon examination of the estimated monthly irrigation rates from the seasonal fields, a systematic mismatch was observed during periods where fields were identified as being "inactive". From knowledge of the local farm-based operations, it is not unusual that fields required a significant amount of pre-planting irrigation (likely to reduce the salt load in the soil and in preparation for seeding). But during this period, the LAI would be below the threshold used to define an "active" season (Section 3.3), and thus water
accounting in the model would not be triggered. Lowering the LAI threshold would not help in identifying this pre-planting stage, because it is already essentially zero when vegetation is not present. Reducing it further to arbitrarily low values would defeat the purpose of seasonal activity detection. As an alternative, the season can be extended to include a pre-planting stage of a pre-defined amount of time. However, this parameter would depend on a variety of factors, such as the type of crop being planted and other farm management practices. Without a sufficient amount of data with which to derive strategies to account
for irrigation during this pre-planting stage, the groundwater abstraction values for seasonal fields in this study can only be interpreted as a lower bound.

Figure 5 presents the groundwater abstraction estimates calculated for the center-pivot fields compared against farm reported flow rates (multiplied by irrigation duration). As can be seen, there is a significant amount of variability between the pivot results, largely a consequence of seasonal versus perennial fields. However, for a number of the seasonal fields, there was a clear and defined under-estimate of groundwater abstraction relative to the reported flow-rate extrapolations (identified by the gray-dots). Indeed, eleven fields had an unrealistically long pre-planting stage of more than two months, based on the reported flow-rate data. It is extremely unlikely that these reflect real farming practices, and are almost certainly a result of local reporting errors. Taking into account the uncertainty of the farm data for these eleven pivots, the yearly groundwater abstraction values were re-calculated to estimate only when fields were determined to be active, i.e. based on the satellite-derived LAI values. Using this assessment threshold, the NSE value was 0.38 with an $R^2$ of 0.61 for all 40 pivot fields, with a linear regression described by the blue line in Figure 5 (slope of 0.62, intercept of 0.51). For reference, the eleven gray dots (not included in the NSE calculation) present the original data that includes the "inactive" period, i.e. with the long pre-planting stage.

Further exploring the relationship between monthly groundwater abstraction and its relation to the level of a field's activity (based on LAI data), Fig. 6 shows the results for two different fields and their spatial and temporal changes in LAI. The first field (center panel) presents a 6 month period when LAI is above the defined threshold (0.3 $m^2/m^2$), and a long inactive period (August – December) which would be designated as low-to-no estimated irrigation. For the period of activity (Jan-July, excluding a planting stage in March) the agreement between reported flow rate and estimated groundwater abstraction is good. However, for the August-December period, when the LAI imagery shows low LAI to bare soil conditions, the farm data reports a varying amount of irrigation based on the fixed flow rate records. For the second field (right panel of Fig. 6), the monthly variations in groundwater abstraction are in good agreement with farm data for the majority of the year, with the exception of a three-month period (January to March), which was identified as inactive, but where farm-reported data again indicates irrigation is active.

While undertaking a field-based assessment of the strategy is obviously an area of importance, it is tempered by the reality of using data that is often of less than "high-quality". However, by combining independent observation available from satellite platforms, we can further discriminate these spurious data points, and refine the assessment process. These types of analyses highlight the need for forensic assessment of ground-based (and satellite) observations, and the value of establishing consistency between available datasets (McCabe et al. 2008; Lopez et al. 2017). Indeed, in this case, the proposed model-satellite fusion approach correctly identifies errors in "ground-based" data, and provides a clear example of the observation and monitoring challenge in this and many other regions.

**4.2 Irrigation activity detection in Al Jawf**

Given the quite different application scale of the pivot-based groundwater estimation approach when performed over the Al Jawf region (compared to Tawdeehiya), an overview and analysis of intermediate processing steps is warranted. The object-based image analysis approach (see Section 3.2) produced a map containing 5,567 individually delineated fields,

covering a total area of 2,494 km$^2$ i.e. more than 60% higher than previous reporting (FAO, 2013). The majority of the delineated fields (81%) were identified as "perennial" (see Fig. 4b), with less than 1.5% recognised as inactive using the LAI-based approach. Examples of these delineations are shown in Fig. 1 (top; outlined in black). On average, the size of the fields were 45 ha (0.45 km$^2$), with no observable distinction in acreage between perennial and seasonal fields. Figure 7 displays the

distribution of size (in ha) for all fields (first quartile of 25.54 ha, third quartile of 66.05 ha). The field sizes do not follow a normal distribution, but form clusters (e.g. around 82 ha, 67 ha, and 50 ha; see inset on Fig. 7), which is expected as center-pivot irrigation systems are installed in standard sizes. The largest fields (e.g. > 60 ha), were concentrated around the central region (30° N, 38.25° E), which is where the largest commercial-scale farms operate. In more remote areas to the north and east of Al Jawf, a larger variation of smaller fields can be identified, and are likely owned by smaller, independent farms.

The annual field use (%), calculated as the ratio between the number of active days and total days in the year, is shown in Fig. 8. As expected, perennial fields have a higher field use (average, first and third quartiles: 86, 77 and 100%, respectively) than seasonal fields (average, first and third quartiles: 35, 24 and 46%, respectively). In contrast with the area distribution (Fig. 7), the majority of fields had high annual field values, meaning they were active throughout most of the year, independent of their location (small or large commercial-scale farms). This is consistent with the fact that most fields were identified as

perennial, and indicates a preference towards forage crops (i.e. grass and alfalfa) during this year, regardless of the scale of operation.

The field use (%) was also calculated on a monthly basis as the ratio of number of days active to number of days in each month in Fig. 9, with the distribution of values among all fields shown as violin plots for perennial and seasonal fields separately. Most perennial fields were consistently active throughout the year (monthly field use above 80%), but for any given

month there were fields that were inactive (i.e. with less than 5% monthly use). For example, the largest number of inactive perennial fields was 958 in January, followed by 913 in December, with the smallest being 353 in March. Because perennial fields are, by definition, expected to be active throughout most of the year, the violin plots (Fig. 9) show a consistent pattern with a wide top (i.e. high y-values) and otherwise thin body. Seasonal fields on the other hand, had a more variable range of irrigation activity. Most fields were irrigated from February to May, with more than 46% of the fields having a monthly field

use value larger than 50%. This was followed by July and August, where more than 33% of fields exceeded the 50% monthly field use. Irrigation activities were lowest during January, June and from September to December, where less than 30% of fields had monthly field use values larger than 50%. This suggests an overall trend for seasonal crops either being used for one growing season (from February to May), or two seasons (February to May and July to August).

### 4.3 Monthly and annual irrigation and groundwater abstraction

The irrigation rates for each field were determined using the inverse modeling approach, i.e. running the CABLE model iteratively to determine the rate needed to reproduce the TSEB-based satellite-observed crop water use (see Section 3). Figure 10 presents the derived monthly irrigation estimates. Results were generally higher for perennial fields compared to seasonal fields, with average values ranging from 122– 152 mm in January, February and from October to December, and the

highest values occurring from April to September (210 – 234 mm). The monthly maximum value for a perennial field was 407 mm and occurred in August. Average yearly irrigation values for the perennial fields reached 2,007 mm, with first and third quartile values of 1,347 mm and 2,799 mm for the fields within the Al Jawf region. Average monthly irrigation values for seasonal fields were lowest in November and December (50 – 58 mm) and highest during March to April (165 – 171 mm), indicating the production of spring crops. A second peak, indicative of a second season (as mentioned in the monthly field use analysis), was also evident in the monthly irrigation profile, with average values of 145 mm and 131 mm in July and August, following a lower value of 109 mm in June. The highest monthly irrigation value for a seasonal field was 348 mm during July. On average, seasonal fields had a total annual irrigation value of 675 mm, with the first and third quartiles at 299 mm and 1041 mm.

The irrigation values were converted to groundwater abstraction by multiplying monthly irrigation amount by the area covered by pixels that were actively irrigated in each season, and then by a factor of 1.25 (i.e. 1/(1-Closs)) to account for irrigation losses (as described in Section 3). Figure 11 shows a map of annual groundwater abstraction in the region, in units of million cubic meters (MCM). The total annual groundwater consumption in 2015 for the Al Jawf agricultural region was estimated at 5.56 billion cubic meters (BCM). Clusters of fields with high abstraction (> 3 MCM/year; shown in blue) are mostly centered in the main commercial region (38.25° E, 30° N), where fields were generally larger (> 60 ha; Fig. 7) and irrigated throughout most of the year (>80%; Fig. 8). The first quartile, mean and third quartile of groundwater abstraction among all fields in Al Jawf was 0.24, 1.0 and 1.59 MCM, respectively. The corresponding values were larger, as expected, for perennial fields: 0.43, 1.16, and 1.74 MCM, and significantly smaller for seasonal fields: 0.06, 0.34, and 0.5 MCM.

## 5 Discussion

### 5.1 Historical efforts towards quantifying agriculture water use

Quantifying the water use of individual agricultural fields has been a research objective for many decades (Jackson et al., 1987; Rana and Katerji, 2000; Kalma et al., 2008), with numerous efforts directed towards improving process-based modeling approaches to characterise the evaporative response (Ershadi et al., 2014; Anderson et al., 2018; McCabe et al., 2019). The challenge has often been related to the availability of data with sufficient spatial and temporal resolution to observe fields in adequate detail, as well as a lack of knowledge on the dynamics of the underlying crop type and condition. In addition, most efforts have tended to focus on monitoring for relatively short periods, or perhaps a single growing season, rather than providing a basis for long-term retrospective or ongoing monitoring. Recent developments exploiting constellations of CubeSats have enabled high resolution in space and time retrievals of key land surface parameters (Houborg and McCabe 2018c), providing enhanced estimates of crop water use and crop development and overcoming the spatiotemporal constraint (Aragon et al., 2018; McCabe et al. 2017b). However, while crop water use is an important variable in delivering insights into water allocation and management, regulators are often most interested in determining the source and volumes of water actually being extracted from reservoirs to supply this agricultural need, not just the net use. This has represented a much more

challenging task, as in-situ data on these systems is often non-existent, and not easily inferred through remote measurement: at least at the scales at which local and regional management needs to be performed.

In many regions of the world, existing or historical groundwater monitoring networks help to inform regional groundwater depletion trends (Shamsudduha et al., 2012; Scanlon et al., 2012; Zhou et al., 2013) and offer insights into related

environmental impacts (Lee and Song, 2007; Erban et al., 2014). Satellite-based gravimetry measurements from GRACE have informed on water storage depletion trends around the world (Rodell et al., 2018), with particular benefit to data scarce regions where the quantification of aquifer depletion would not otherwise be possible (Lezzaik and Milewski, 2018). However, while GRACE data provide an excellent source of large scale information on aquifer response (Voss et al., 2013; Famiglietti et al., 2014), it is not suited to attribute to any particular use at the scales required for resource management. For example, the Al

Jawf agricultural region as defined in this study (mapped agricultural area of about 2,500 $km^2$) is small compared to the scale of the Saq aquifer system that feeds it (about 500,000 $km^2$) and the recommended minimal size for GRACE studies (>200,000 $km^2$; Famiglietti et al., 2014; Long et al., 2015; Richey et al., 2015). Moreover, even in regions where groundwater monitoring networks do exist, there is a need to bridge the gap between a regional assessment and practical farm scale monitoring. In this context, our study demonstrates new capability, using a satellite data-modeling framework to provide an unprecedented level

of information for water management. The approach represents a dramatic improvement on more traditional farmer-based surveys, which are time-consuming to collect, can often be unreliable and unrepresentative, and lack the spatial and temporal detail needed to provide accurate water accounting at the regional-scale.

While a number of studies have attempted to estimate irrigation by incorporating an irrigation module into a water balance model, these approaches have often been based on "adding" the necessary irrigation depth to maintain the soil moisture

above a threshold value (Santos et al., 2008; Ozdogan et al., 2010, Pokrhel et al., 2012), which may not reflect the actual irrigation volume being applied. Here we developed a data-modeling approach to automatically retrieve seasonal irrigation rates for individual center-pivot fields, focusing on fields irrigated by center-pivot infrastructure: consistent with the type of infrastructure that supports the majority of irrigated fields in Saudi Arabia and in other cereal crop production areas world-wide. The developed approach is based on constraining a land surface hydrology model with evaporation estimates, and then

"inferring" the irrigation rate: an idea explored conceptually by Droogers et al. (2010) and applied in a real-world case-study by Lopez (2018). As the first large-scale demonstration of this framework at the regional-scale, the present study represents an effort towards more effective water use monitoring in both Saudi Arabia and other arid countries (e.g. in the MENA region) that rely heavily on groundwater abstraction for agricultural production.

## 5.2 Demonstrating the capability of the pivot-based groundwater abstraction estimation framework

The framework offers a unique monitoring and modeling effort in terms of scale and granularity, as it demonstrated a capacity to obtain agricultural water use estimates from the scale of a single pivot to more than 5,000 individual fields. Importantly, the approach is scalable and can be applied to other domains and locations. The mapping activities used in producing our water use estimates for the year 2015 indicate a much larger extent of agriculture (2,494 $km^2$) in the region than

has previously been reported (1,200 km$^2$ for the year 2009; SSYB, 2010). Consequently, a much higher groundwater abstraction was also estimated (5.56 BCM) than that forecast for the year 2014 (1.2 BCM; MEP, 2010). However, it is important to note that these prior estimates were obtained by incorporating information from various private and public sources, including on-site interviews, and are subject to significant uncertainties. These include possible misrepresentation due to the absence of metering in farms (i.e. self under reporting), and possible omission of fields located in remote areas. Regardless, our estimates are proportionally consistent (in terms of area) with more recent reports of nationwide groundwater abstraction for the year 2015 (20.8 BCM; MEWA, 2019). That is, both the estimates of area and of groundwater abstraction for the Al Jawf region represent about one quarter of irrigated agriculture in the Kingdom.

Ultimately, our study is a demonstration of using the best available data and tools to undertake an analysis in a data-limited region. The lack of data is very much a developing world problem, but there are numerous "developed" world cases where data to inform model set-up and evaluation is absent. Using the strategy proposed here, we were able to provide results that reflect expectations when compared to the limited available datasets (both at farm and regional scale), i.e. a positive outcome given the scenario where independent data is not available to inform decision makers.

## 5.3 Limitations, potential extensions and applications

The goal of this study was to provide a first approximation of regional groundwater abstraction independent from self-reported data, and for this, we have used a specific choice of models (i.e. TSEB and CABLE). Other remote sensing approaches for evaporation estimation could be used. For example, the use of ALEXI/DisALEXI (Norman et al., 2003) could be explored to mitigate the sensitivity to the accuracy of LST retrieval, but the trade-off between higher ET accuracy and the impact on computational effort should be evaluated. Likewise, our approach is not limited to any particular land surface model. Further investigation is required to determine the uncertainties of the models used as well as from other inputs such as LAI and how these propagate through our groundwater abstraction framework. One approach that could help mitigate biases within specific models is to explore the use of multi-model estimates, which would also help provide ranges of groundwater abstraction.

Because this study was aimed to retrieve estimates for the year 2015, a key reference year that will be used to evaluate the impact of policy changes in Saudi Arabia, Landsat 8 imagery was the primary source of satellite data. With a sun-synchronous return frequency of 16 days, this means that for a 90-day season, the method is based on between 4 to 6 images. Given this low number of observations, our study aimed at retrieving seasonal irrigation amounts, i.e. we do not estimate irrigation amounts in different crop developmental stages. Added to this limitation, cloud cover can be an important factor in the uncertainty of the irrigation activity detection. While this is not a major issue in the current setting (high percentage of "blue-sky" days throughout the year), it may be pertinent to applications in other geographic locations. For more recent years, data from newer satellites with higher temporal resolution, such as Sentinel-2 (Ferrant et al., 2017; Veloso et al., 2017) and CubeSats (McCabe et al., 2017b; Houborg and McCabe, 2018b), could be employed to support improved estimates, as active irrigation seasons would be more accurately defined, and irrigation estimates could be obtained on a sub-seasonal basis (e.g.

being able to differentiate between crop stages). The higher spatial resolution available (e.g. 3 m in the case of Planet data) can also benefit the model, especially for the detection of partially irrigated, or partitioned (two-crop) fields, and to avoid the edge overlap in some fields that hinders the automated capability of the delineation procedure. However, the increased computational cost of using higher resolution data, particularly with the LSM runs, is an important consideration (Wood et al.,

2011; Bierkens et al., 2015), especially when seeking to apply the methodology over a larger number of fields (e.g. at a national-scale).

Detecting periods of active irrigation is a crucial step of the framework, and ensures that the model is able to retrieve irrigation values for the appropriate periods. Ideally, the more often that satellite information is available, the better the prediction of active irrigation seasons will become. This is especially important for perennial fields, which undergo fast

response to cut and re-growth cycles that could be missed by the 16 day revisit interval of Landsat (Houborg and McCabe, 2018b). However, an aspect that requires further investigation is how to retrieve irrigation rates during the pre-planting stage. One way to tackle this challenge would be using high-resolution soil moisture products to track the change in soil water content (Sánchez-Ruiz et al., 2014). Although this was an unexplored aspect in this study, the obtained estimates can be interpreted as a lower bound for seasonal fields, which comprised less than 20% of the fields identified in the Al Jawf region.

As this study focused only on center-pivot irrigation, the total quantity of abstracted water used for agricultural production in Al Jawf will be marginally higher than the reported 5.56 BCM. This value represents a first-order estimate that can be further refined by adding the contribution of other irrigated crops and fields e.g. date palms and more recently agricultural shifts to planting of olive trees. In parallel with additional crop mapping and identification activities, efforts are also being directed towards strategies that monitor water use from other types of irrigation (e.g. drip and flood irrigation), in

order to obtain a more comprehensive estimate of groundwater abstraction in this region and beyond. As a first step, using the object-based image analysis procedure, we delineated and estimated an area of about 31 km$^2$ of irrigated fields in the Al Jawf region that were not classified as center-pivot fields. This represents about 1.22 % of the total irrigated area, and depending on the type of irrigation (e.g. drip irrigation), represents a relatively small fraction of the total groundwater abstraction. One approach to incorporate the agricultural water use from these remaining fields would be to first implement an advanced crop

classification scheme (Cai et al., 2018; Piedelobo et al., 2019) and then calculate irrigation requirements for each crop (Castaño et al., 2010; Kirby et al., 2013; Yang et al., 2019). The reason for using other approaches with these remaining irrigated fields, is that the framework relies on the assumption of relatively uniform irrigation application, which simplified the effort to translate irrigation rates into abstracted groundwater. Additional improvements to the methodology would also include better quantifying the spray-loss component of the center-pivot system. For example, information derived from wind speed and

direction, humidity and air temperature could be used to refine spray loss estimates over each field (Abo-Ghobar 1992; Sadhegi et al., 2015; 2017) with a dynamic, rather than fixed value. The methodology is being further developed to run both retrospectively and up to current-periods in order to monitor change in agricultural activities across the Kingdom and to quantify this sectors corresponding water use. Such data will enable responses to policy changes and management

implementations to be identified, and can act to facilitate optimization practices for agricultural water use and groundwater abstraction in other data-sparse regions.

## 6 Conclusions

An automated approach to estimate agricultural-driven groundwater abstraction based on integrating satellite data and land surface modeling was developed, with its functionality demonstrated over several thousand center-pivot fields in an arid region of Saudi Arabia. The monitoring framework provided an unprecedented level of information capturing water-use behaviour at the individual field-scale, and included information metrics such as geospatial location, distribution of cultivated areas, irrigation activity patterns, crop water use, and ultimately the abstracted groundwater used to grow the agriculture product. Monthly to yearly estimates of abstracted groundwater were obtained for more than 5,000 center-pivot fields, covering an area of approximately 2,500 km$^2$, with a total groundwater use estimated at 5.56 BCM. Individual field use ranged from 0.24 MCM to more than 3 MCM annual abstractions for those areas operating at more than 80% capacity. The annual total abstraction value represents about one quarter of the total groundwater abstraction used for agriculture in the Kingdom (20.8 BCM; MEWA, 2019). In terms of agricultural area, the 2,500 km$^2$ also represent about one quarter of the Kingdom's center-pivot based irrigation capacity (~10,029 km$^2$; FAO, 2008a).

With the development of this novel water accounting approach, changes and trends in agricultural patterns from regional to national scales can now be monitored, providing information on crop type (perennial or seasonal), changes in the cultivated areas, and volumes of water being used over time. Such information is needed for improved water management, to inform the development of water related regulations, and to assess the impact of policies on water conservation. The approach is currently being deployed retrospectively to monitor all center-pivot infrastructure across Saudi Arabia for the years 2011–2015, and then to expand this forward in time to allow near real-time monitoring. Future work will focus on the inclusion of other types of agriculture (e.g. date palms, orchards and olive trees) for a more complete accounting of water abstraction for agricultural use. In parallel, a classification of crop types grown within individual center-pivot fields is being performed to better identify potential water-saving and irrigation optimization techniques at the individual field-scale. The availability of new and emerging sources of remote sensing information presents an opportunity to further advance our precision agricultural capacity, and will be incorporated into the future versions of this modeling framework, providing enhanced assessment on crop growth and field condition.

## 7 Data and code availability

Landsat 8 imagery used in this study is publicly available from the Google Cloud Platform at https://cloud.google.com/storage/docs/public-datasets/landsat. Data from the WRF reanalysis performed within this study is available upon request. The source code of CABLE version 2.3.4 used in this study is available at *trac.nci.org.au/trac/cable*

after registration to the CABLE user group at the National Computational Infrastructure (NCI) Australia. Python code to run the TSEB model and the pivot-based groundwater abstraction strategy used in this study is available upon request at hydrology@kaust.edu.sa.

## 8 Author contribution

OL and MFM designed the pivot-based groundwater abstraction methodology, with advice on model inversion conceptualization by UA. KJ designed and applied the semi-automated delineation of center-pivot fields. BA implemented the TSEB model using Landsat data. TL designed and applied the approach for irrigation activity detection in center-pivot fields. RH applied the REGFLEC model using Landsat and ancillary data for the retrieval of vegetation indices and biophysical parameters. YM applied the atmospheric correction and sharpening of LST data. HPD and IH applied the WRF model to obtain the meteorological data needed to drive the TSEB and CABLE models. EMF and SM obtained and curated data for evaluating the model performance. OL prepared the manuscript with contributions from all co-authors.

## 9 Competing interests

The authors declare that they have no conflict of interest.

## Acknowledgements

Research reported in this publication was supported by the King Abdullah University of Science and Technology (KAUST).

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

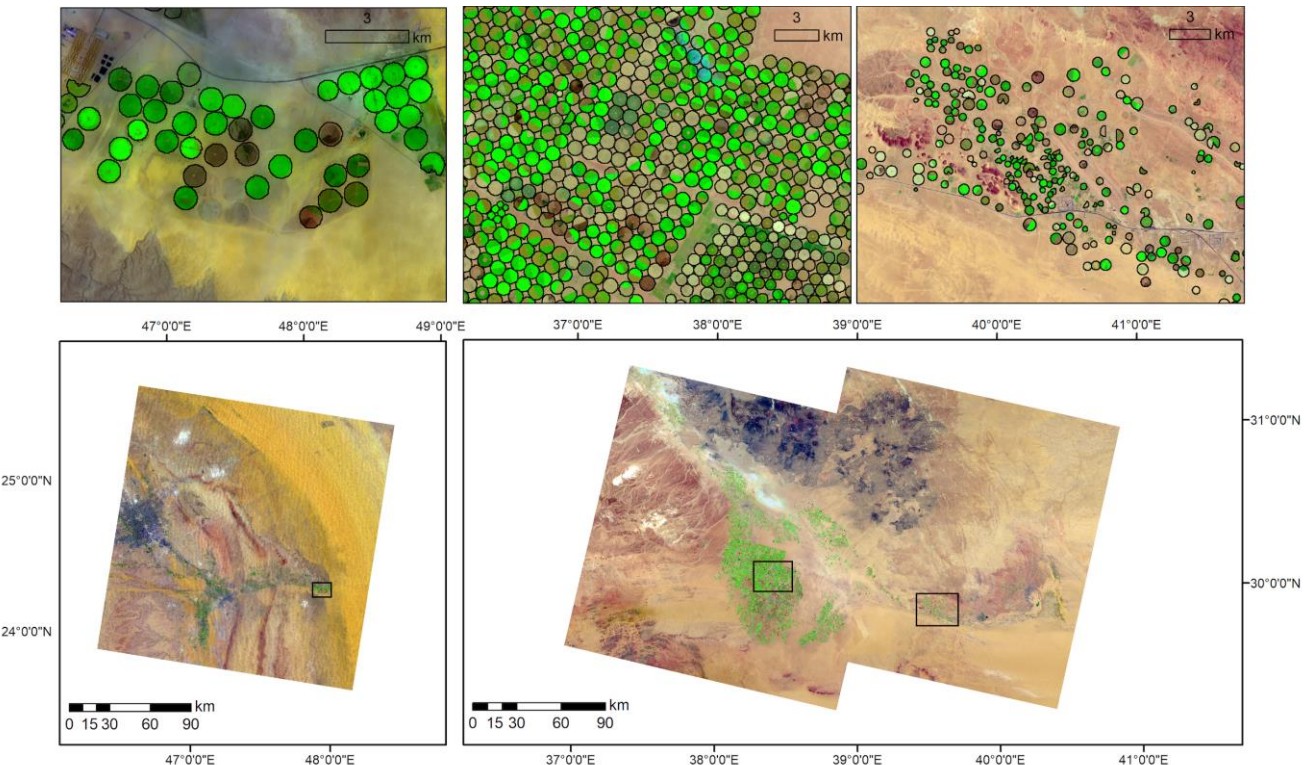

**Figure 1: Location of the two study regions. Left: The Tawdeehiya farm in Al Kharj (southeast of Riyadh). A false color Landsat 8 image (2015/06/09) is shown to highlight active center-pivot fields over the desert environment. Right: The Al Jawf agricultural region in the north-west of Saudi Arabia spans two Landsat 8 tiles. Two false color images are shown: 2015/06/09 for path/row 172/39 (left) and 2015/06/19 for path/row 171/39 (right). Center-pivot fields are densely packed and largely uniform in size in the main area (30° N, 38.25° E), while in other areas they are sparser and less uniform (for example, the image on the right).**

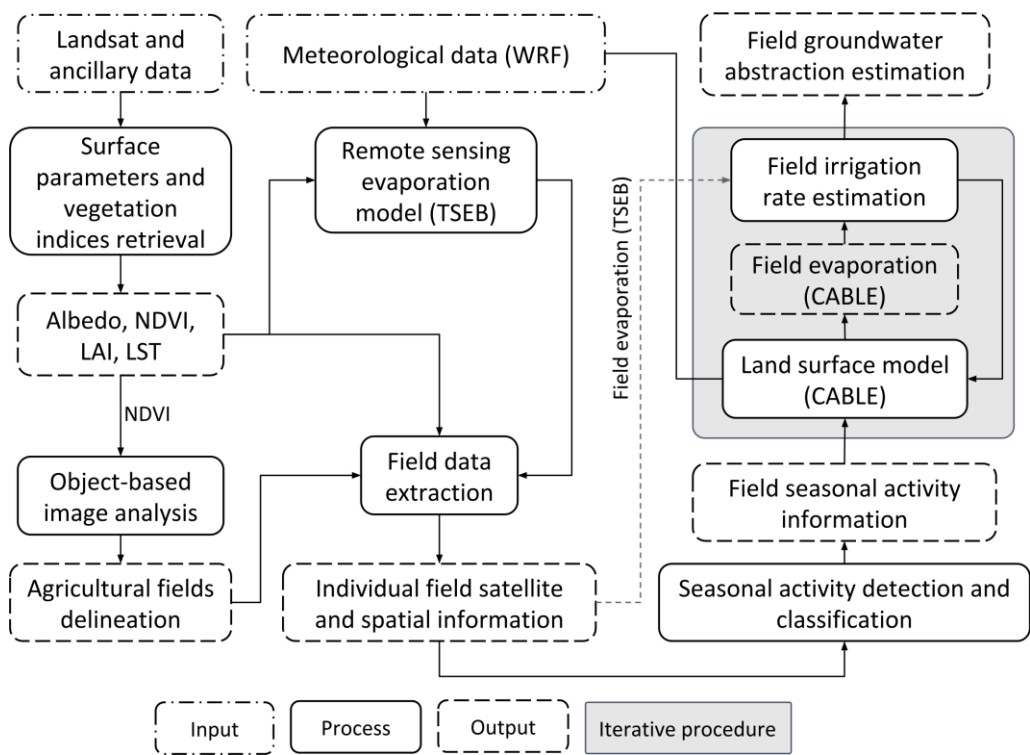

**Figure 2: Flow chart outlining the main inputs (dashed-dotted), processes (dot pattern) and outputs (solid) of the groundwater abstraction monitoring framework.**

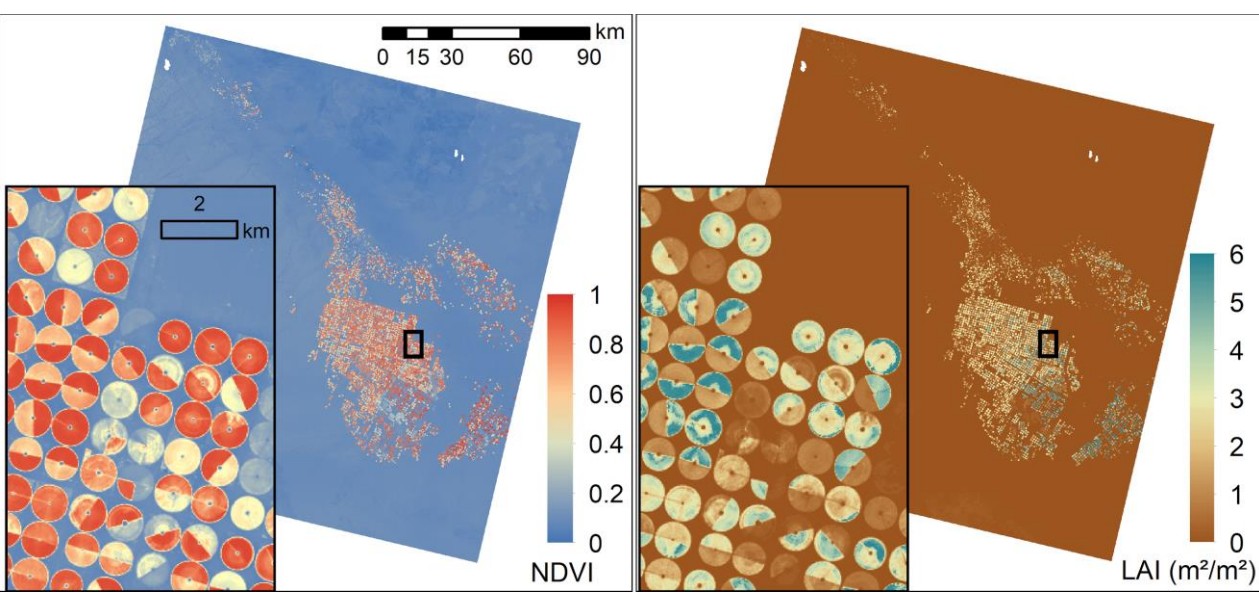

5    **Figure 3: Example of the full Landsat tile (path/row 172/39) NDVI (left) and LAI (right) estimation, demonstrating the foot-print from center-pivot irrigated fields in this region (high contrast with the bare desert soil).**

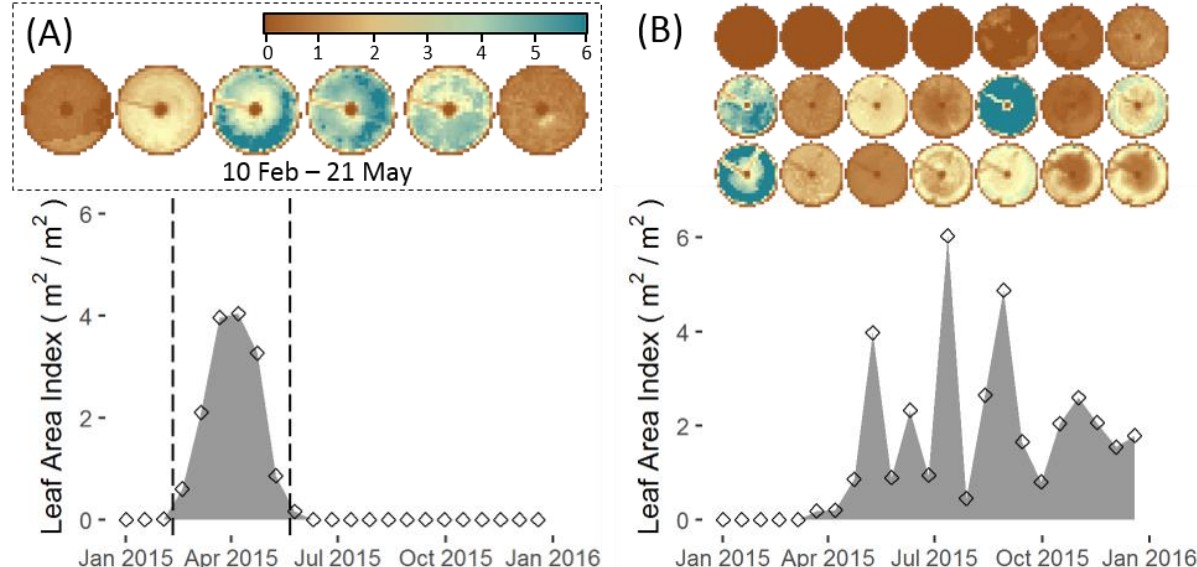

**Figure 4: Example of two types of crops identified for this study based on an LAI threshold of 0.3 m²/m². (A)** Images of the six scenes when the field was identified as active are shown on top, which correspond to the six dates marked as diamonds inside the dashed lines on the bottom plot. The season start and end dates (10 Feb and 21 May) correspond to dates when the mean LAI crosses the threshold of 0.3 m²/m². **(B)** This particular field was inactive during the first three months of the year, followed by large LAI oscillations, indicating repeat cut/re-growth activities. Landsat scenes for this field are shown on top, increasing in date to the right and bottom, while the dates are marked as diamonds in the LAI time series plot.

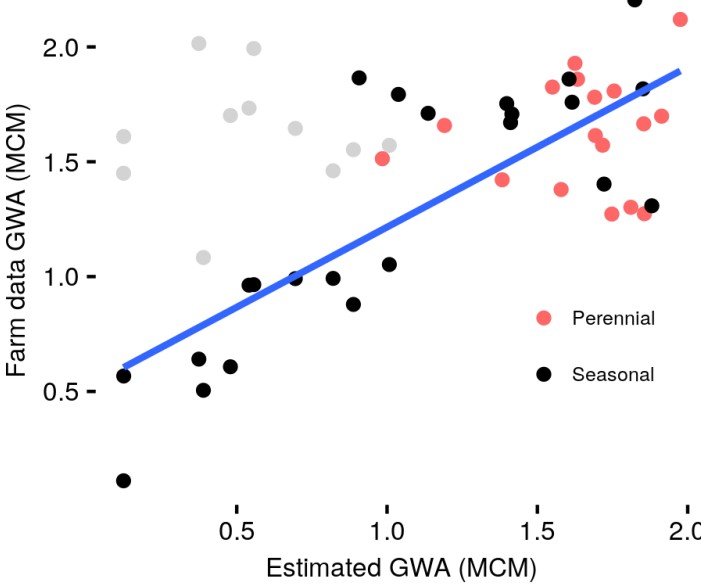

**Figure 5: Comparison of annual groundwater estimates to farm data.** The blue line shows the regression based on the black and green dots (adjusted to include only active seasons for 11 seasonal fields). The gray points show the original farm data with long pre-planting stages for those same 11 seasonal fields.

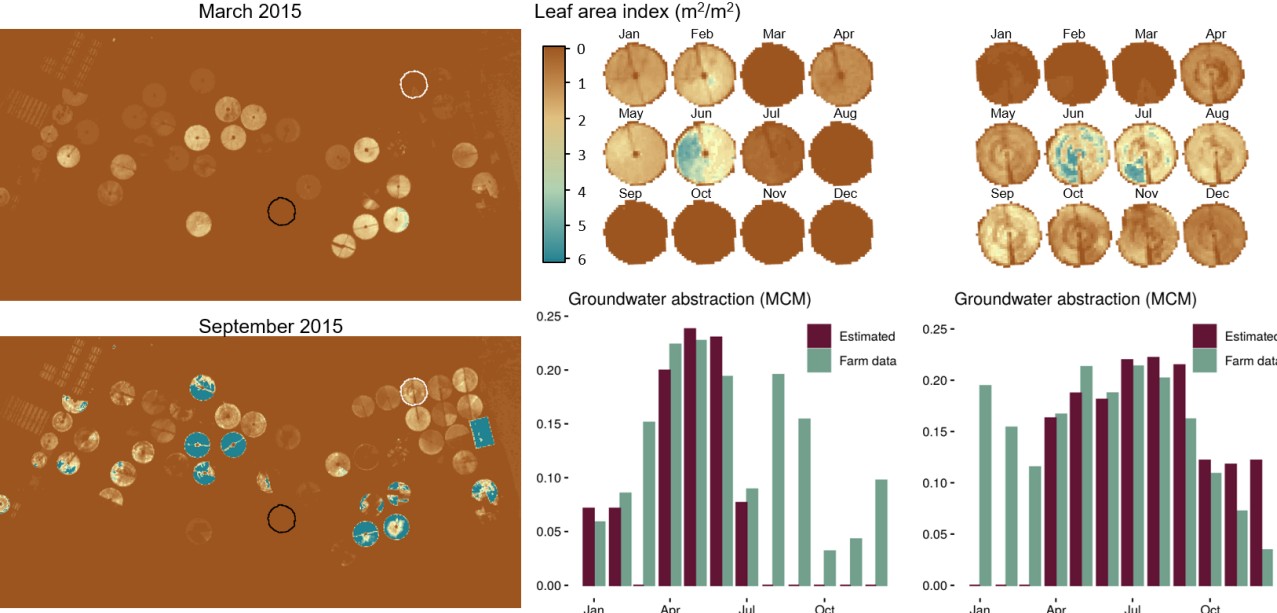

**Figure 6: Estimated groundwater abstraction in million cubic meters (MCM) for two fields in the Tawdeehiya farm along with a comparison based on available flow rates from farm data. The left panel shows the spatial maps of LAI data ($m^2/m^2$) using the methodology described in Section 3.1. Two fields from two different periods are delineated either in black (corresponding to middle panel) or white (right panel). Each of these panels shows a spatiotemporal map of the field LAI (top), and a comparison of groundwater abstraction obtained using the framework described in Section 3 and available farm data (bottom). The field marked with black is one of the eleven fields identified as having a large abstraction discrepancy i.e. the farm data indicates ongoing periods of irrigation, while the LAI data indicates inactivity.**

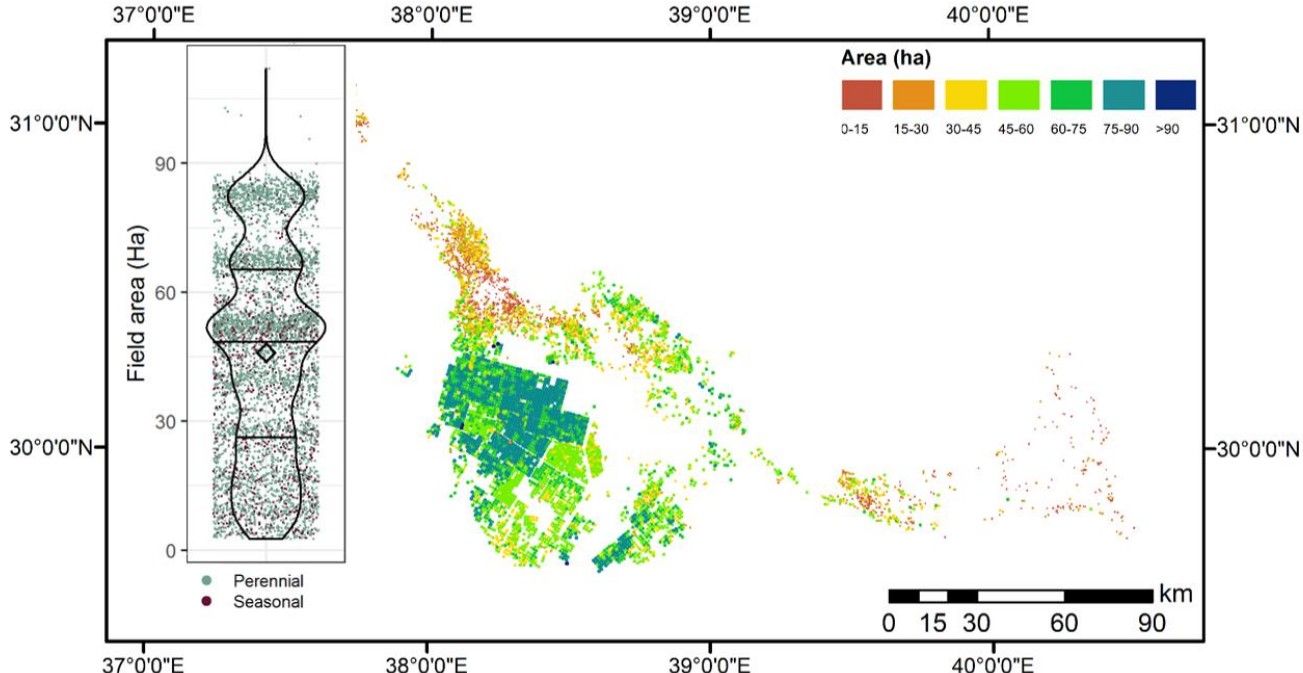

**Figure 7: Spatial distribution and statistics of individual field areas. Clusters of large fields (those > 60 ha) can be found in the main agricultural zone (30° N, 38.25° E), corresponding to the location of several large commercial-scale farms. The figure's inset on the left shows a violin plot of the field sizes in Al Jawf: at a given field area (y-axis), the plot outline (in black) is wider when there is a larger number of fields of that given size. The black horizontal lines inside the plot show the first quartile, median and third quartile, while the black diamond show the average value. The background on the inset shows colored dots (horizontal positions are given randomly for visualization purposes) to distinguish perennial and seasonal fields.**

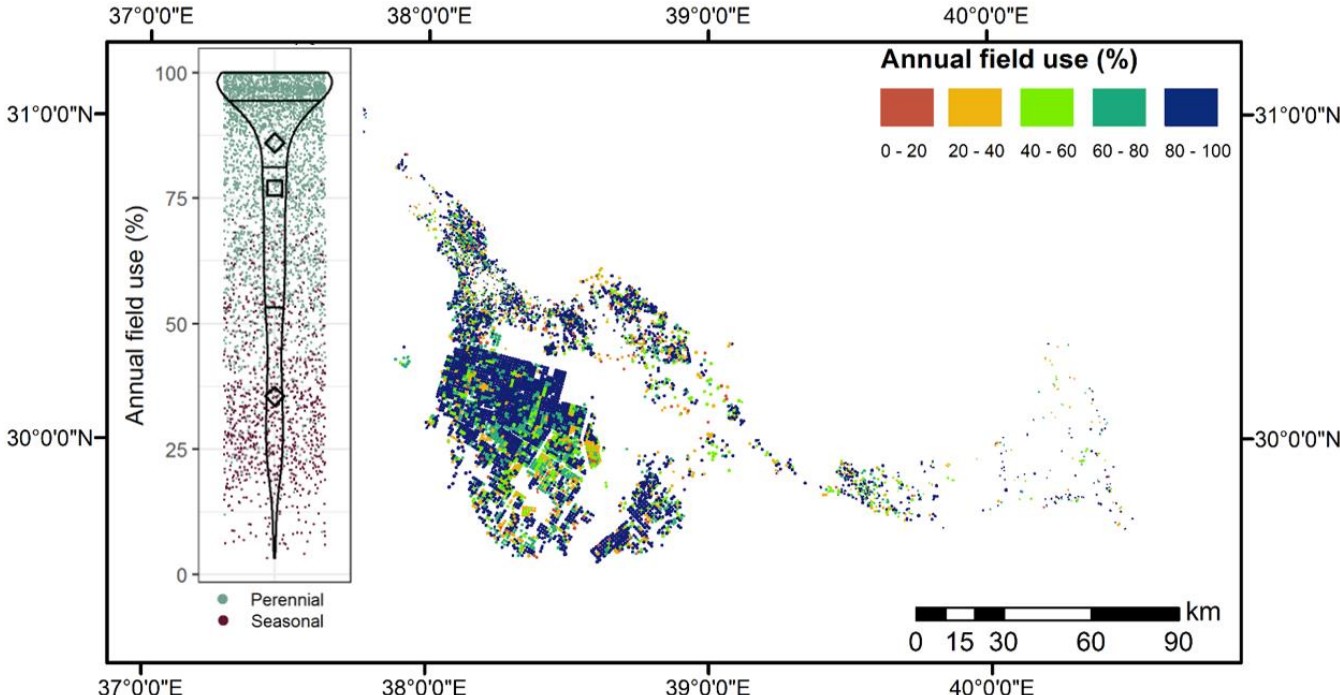

**Figure 8: Spatial distribution and statistics of the annual field use, defined as the ratio of active irrigation days to total number of days in the year. Most fields had high values of annual use (> 80%). The inset on the left shows the distribution of annual field use among all fields in the region as a violin plot. The two black diamonds show the average value grouped by the type of field (seasonal or perennial), while the black square represents the average of all fields. The background shows colored dots (horizontal positions are given randomly for visualization purposes) to distinguish perennial and seasonal fields.**

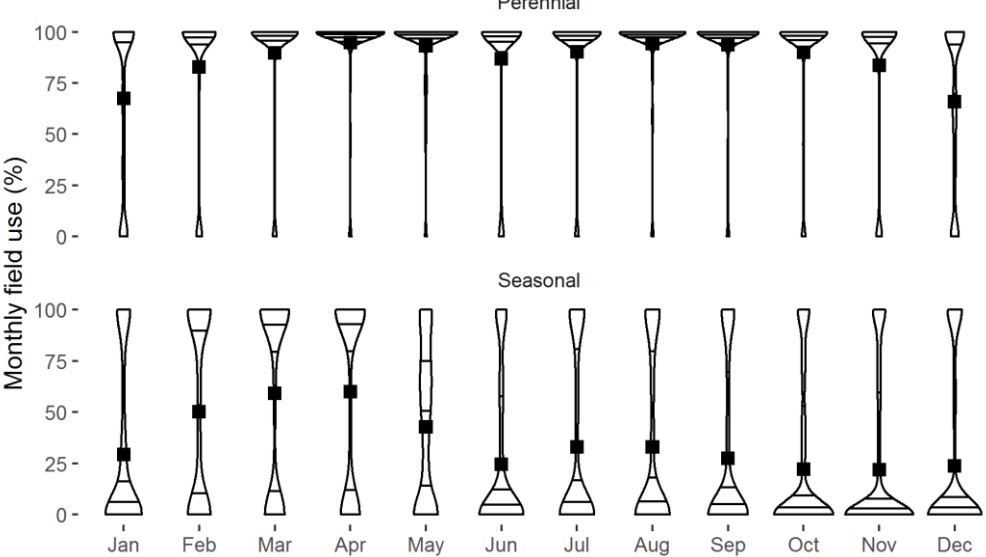

**Figure 9: Monthly center-pivot field use for perennial (top) and seasonal (bottom) fields. The black squares show the average monthly field use (% days irrigated during each month). A larger width at a given level of use indicates a larger number of fields. Horizontal lines show the 25%, 50% (median) and 75% quantiles.**

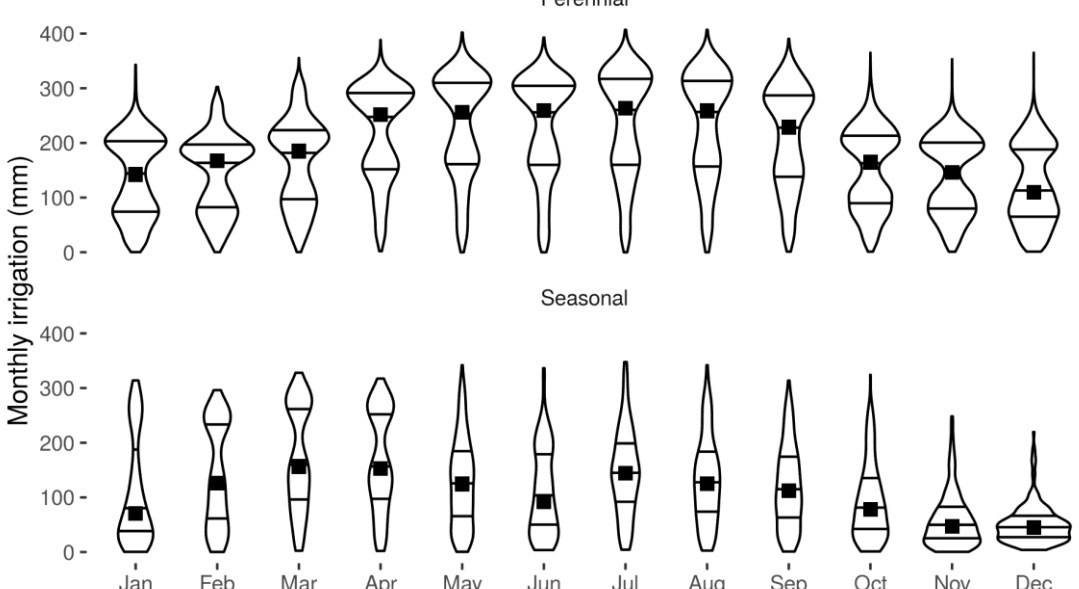

**Figure 10: Monthly irrigation statistics (mm) for perennial (top) and seasonal (bottom) fields. The black squares show the average values of monthly irrigation among the same type of fields in the region (4509 perennial fields; 974 seasonal fields). The violin plots show the distribution of monthly irrigation values, which range from 0 (no irrigation) up to 406 mm/month (i.e. 13.5 mm/day). Horizontal lines show the 25%, 50% (median) and 75% quantiles.**

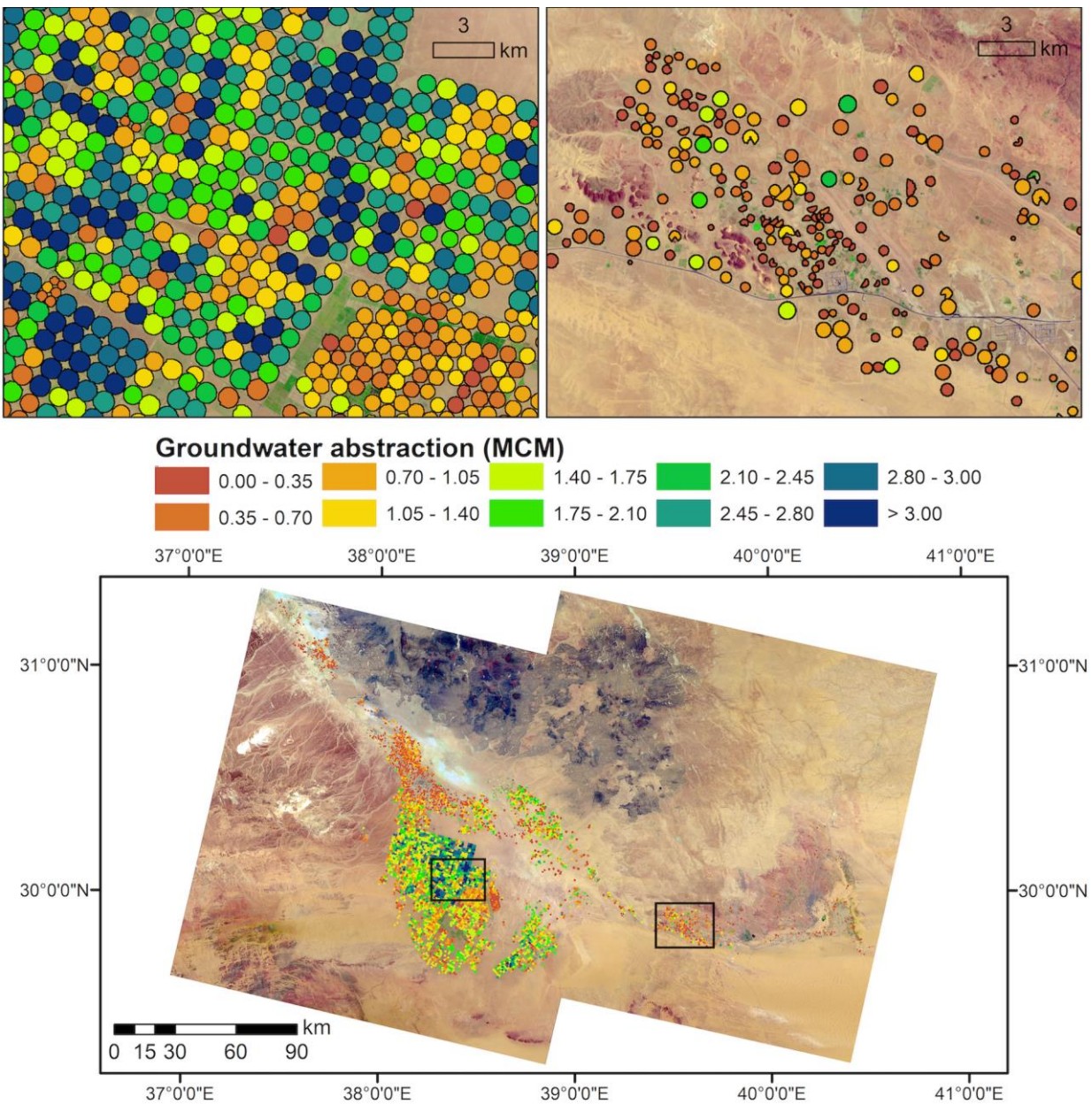

**Figure 11: Map of annual groundwater abstraction in million cubic meters (MCM) for the Al Jawf agricultural region. Values were obtained for individual fields, as seen on the examples shown at top featuring one zone with high levels of abstractions (top-left) and another zone with a smaller density of fields and lower values of abstraction (top-right). The background shows the same Landsat 8 images as in Fig. 2.**