# Peer review of "Mapping groundwater abstractions from irrigated agriculture: big data, inverse modeling and a satellite-model fusion approach"

_Hydrology and Earth System Sciences, 2020_

## Referee Comment (RC1) · Anonymous Referee #1 · 12 Apr 2020

This study develops an automated approach to estimate agricultural-driven groundwater abstraction by integrating satellite data and land surface modeling. While this research is important and the groundwater estimation results look promising, the methodology is not well elaborated and justified.

1. The land surface model CABLE is the core of the methodology. Only if CABLE could correctly simulate each horological component, the water input could be inferred from ET. This is a big concern as land surface models are known to overestimate the coupling between soil moisture and ET (10.1029/2018WR023469). How the model is calibrated in the study area? What's the sensitivity of ET to water input? What's the

model performance in forward simulation? What's the underlying uncertainty of model inversion? To what degree TSEB ET improve CABLE ET? All these questions are not addressed and therefore the methodology is not convincing, even though the final results look good.

2. ET is the link between the land surface model and satellite remote sensing. However, no ET result is shown in the manuscript at all, and there's no comparison between TSEB ET, CABLE ET and field observation. While TSEB is a reasonable model, it is sensitive to the accuracy of LST but LST retrieval is usually error-prone (that's why ALEXI is developed to mitigate this issue), in particular for the TIRS on aboard Landsat. TSEB is also sensitive to some configurations such as the selection of resistance scheme. The authors use "in-series" scheme while other papers use "in-parallel" scheme. How these intermediate steps were determined and how they perform worth demonstration.

3. The authors shows that the methodology is able to apply to large scale. While this is cool, I wonder if it is possible to evaluate the water budget by e.g., comparing with GRACE or other horological data? I'm still worried that water loss other than ET may not well captured by CABLE and may couse large uncertainty on the relationship between water input and ET.

P3L14: Landsat (7 & 8) is 8 days. P9L28: Why not using Houborg's approach as I note he is in the author list. P9L30: Need to specify the configuration of the training database derived from PROSAIL. P13L14: Why series scheme is chosen as there are studies recommend parallel scheme. P13L19: Is canopy height involved in the calculation? If so, how do you deal with that? P13L26: How do you calculate roughness length without sensible heat? P13L29: Need a citation or justification. While shortwave radiation may follow Beer's Law (and there is impact of leaf angle distribution), longwave radiation is different (you may check CABLE's longwave radiation radiative transfer). P14L8: This assumes a spherical leaf angle distribution. Need elaboration. P14L9: Then is there a sharp change when LAI=1 as you use two different models? P14L16:

[Figure]

Please specify parameter values. P14L20: How is albedo calculated? P14L27: Isn't CABLE a dynamic vegetation model, i.e., LAI is calcualted in a process? How could you use LAI as input which breaks the process? P15L5: Is there a dynamic root consideration? P15L13: How could you figure out these soil parameters? P15L16: I believe there are a large amount of parameters in CABLE which can influence ET response to water input. Do you do any calibration and sensitivity analysis?

---

## Referee Comment (RC2) · Anonymous Referee #2 · 4 May 2020

This manuscript presented an interesting framework to estimate groundwater abstractions for agricultural irrigation. The framework is novel as it integrate land surface modeling, satellite remote sensing, and inverse modeling. This work was built upon several other works from the same group of people. The manuscript is well written, though the organization part can be further improved. I have the following concerns for the authors to consider:

Firstly, more details about CABLE model simulation should be given. For example, is there a crop model in it? How was the simulation conducted for different crops (maize-C4 and wheat-C3, and others)? What if there are more than two crop types (if that is

possible in this region) within a single field? How was the spin-up conducted? What's the model's performance in simulating ET when giving observed LAI compared with flux-tower or other ground-based measurements?

Secondly, the only validation reported in this manuscript is in Fig. 6 and 7 showing comparison between the annual/seasonal model estimation and farm reported groundwater abstraction. There is no validation on LAI and ET estimation from the satellite remote sensing. If there is uncertainties, how would they propagate to the final estimation of groundwater abstraction?

Thirdly, was the inverse modeling conducted at daily time scale? If it is, are we expecting irrigation every day, which is absolutely not true in the reality? The authors reported the accumulated amount of irrigation water use at monthly and annual time scale. How about irrigation timing and times?

Fourthly, where did the authors assessed the model performance as described in section 3.5?

Finally, organization of this manuscript can be further improved. For example, Fig. 1 and Fig. 2 can be combined? Descriptions of TSEB and CABLE models can be simplified and details about them can be put into supplementary? Instead, the authors should focus more on simulation protocols there.

---

## Author Comment (AC1) · 8 Jun 2020

We thank the reviewer for their comments and questions that have helped to improve the description of the methodology. As we integrate a large number of different elements in this framework, we recognize that some descriptions can be improved for clarity.

**Comment 1**: The land surface model CABLE is the core of the methodology. Only if CABLE could correctly simulate each hydrological component, the water input could be inferred from ET. This is a big concern as land surface models are known to overestimate the coupling between soil moisture and ET (10.1029/2018WR023469).

**Author's response**: We agree that there are challenges in hydrological component estimation with land surface (and even process) models. However, models serve as the best tool for providing an interpretive framework, particularly when used in the context of ingesting or evaluating against available observations. While a particular combination of models were used in the current approach (CABLE and TSEB), based on both past model experiences and their previously evaluated suitability, we do explicitly recommend exploring the use of different models, or even an ensemble of models. Our framework certainly provides this flexibility. To highlight this, we have now included the following text in the Discussion section (page 21, line 6): "The goal of this study was to provide a first approximation of regional groundwater abstraction independent from self-reported data, and for this, we have used a specific combination of models (i.e. TSEB and CABLE). Further investigation is required to determine the uncertainties of these models – as well as other inputs – and how they might propagate through our groundwater abstraction framework. One approach that could help mitigate biases within specific models is to explore the use of multi-model estimates, which would also help provide estimated ranges of groundwater abstraction."

**Comment 2**: How the model is calibrated in the study area? What's the sensitivity of ET to water input? What's the model performance in forward simulation? What's the underlying uncertainty of model inversion? To what degree TSEB ET improve CABLE ET? All these questions are not addressed and therefore the methodology is not convincing, even though the final results look good.

**Author's response**: We have split our response to answer these questions below. However, as a general response to this comment, it is important to note that this study represents a demonstration approach to provide meaningful information in a data poor environment. That is, it was not intended as a calibration/validation effort of a predictive model (or integration of models) framework, but rather as an application that builds on decades of research in the use of remote sensing data and land surface models.

**Comment 2.1**: How is the model calibrated in the study area?

**Author's response**: We did not perform a traditional calibration of parameters in the CABLE model. One reason is that this study is not performed within a defined basin with an observable runoff component, which is the usual approach to model calibration. Absent such observations, we are limited to running the model in default mode. While there have been efforts to undertake multi-observation based calibration approaches (which our team has explored previously; McCabe et al., 2005; Stisen et al., 2011), these are certainly not commonplace: and again run into the problem that one of our observations is actually used in the optimization target (i.e. independently observed ET). Our study "area" is in fact 5000+ individually delineated center-pivot fields, in which we employ the model as a 1D representation to capture vertical moisture transport. That is, we are most interested in using the model physics to capture the underlying hydrological processes related to hydrological partitioning. Therefore, CABLE parameterization was done based on regional soil parameters but with vegetation characteristics determined by satellite observations (e.g. LAI). Using these parameters, we employ the model as an optimization tool to infer the irrigation amount. Even running the model in a "forward" mode is not realistic, as rainfall is non-existent – and without a coupled irrigation module, we developed an approach to infer this variable indirectly.

McCabe, M., Franks, S., Kalma, J. (2005). Calibration of a land surface model using multiple data sets. Journal of Hydrology, 302(1-4), 209-222.

Stisen, S., McCabe, M. F., Refsgaard, J. C., Lerer, S., Butts, M. B. (2011). Model parameter analysis using remotely sensed pattern information in a multi-constraint framework. Journal of Hydrology, 409(1-2), 337-349.

**Comment 2.2**: What's the sensitivity of ET to water input and what's the model performance in forward simulation? To what degree TSEB ET improve CABLE ET?

**Author's response**: It is important to note that a forward simulation of the land surface model cannot be performed, precisely because the irrigation amount is not known (and

rainfall is non-existent). That is, implementing the land surface model using only the limited rainfall amount in this region, leads to an almost complete lack of evaporation signal. On the other hand, TSEB ET is derived from satellite observations, and does not require prescribing the irrigation amount applied. Therefore, TSEB ET is used to infer the irrigation amount that would be needed in CABLE to reproduce the estimated ET amount. In that sense, TSEB ET improves the estimation of CABLE ET by providing the missing irrigation component in this region.

**Comment 2.3**: What's the underlying uncertainty of model inversion?

**Author's response**: In this and many other such regions, ground based collections of irrigation data do not exist for individual fields at a sufficiently large scale to evaluate the uncertainty of model inversion. These and other issues were mentioned in Section 4.1. Ultimately, our study is a demonstration of using the best available data and tools to undertake an analysis in a data-limited region. This is very much a developing world problem, but there are numerous "developed" world cases where data to inform model set-up and evaluation is absent. Using the strategy proposed here, we were able to provide results that reflect expectations when compared to the limited available datasets (both at farm and regional scale), i.e. a positive outcome given the scenario where independent data is not available to inform decision makers.

**Comment 3**: ET is the link between the land surface model and satellite remote sensing. However, no ET result is shown in the manuscript at all, and there's no comparison between TSEB ET, CABLE ET and field observation. While TSEB is a reasonable model, it is sensitive to the accuracy of LST but LST retrieval is usually error-prone (that's why ALEXI is developed to mitigate this issue), in particular for the TIRS onboard Landsat. TSEB is also sensitive to some configurations such as the selection of resistance scheme. The authors use "in-series" scheme while other papers use "in-parallel" scheme. How these intermediate steps were determined and how they perform worth demonstration.

**Author's response**: As the main contribution of this work is to estimate groundwater abstraction, we focused our attention only on this component in the results section. However, we have now added a figure showing ET results to add confidence to the results as supplementary material (Figure S1).

We would note that the potential of TSEB (and other remote sensing models) for estimation of ET over irrigated crops in semi-arid and arid regions has been well documented and validated (e.g. Colaizzi et al., 2012; Zhuang and Wu, 2015; Nieto et al., 2019), including initial efforts from our group to validate TSEB (among other evaporation models) within Saudi Arabia (Aragon et al., 2019). At some point (and after decades of research), the hope is that such remote sensing based approaches can ultimately move beyond being purely research tools, to actually be implemented to provide guidance for operational management. That is an aspect that we explore here.

However, to address the concern, we have also added a figure (Figure S2) showing a comparison of TSEB and measured LE over an irrigated field in one of the study regions (Tawdeehiya farm; see Figure 1). TSEB was chosen to reduce the already high computational load of running CABLE and an ET model over this large region. Nevertheless, we are aware of the sensitivity of TSEB to the accuracy to the LST retrieval, and we are actively exploring the use of ALEXI and its associated disaggregation scheme (DisALEXI; Anderson et al., 2011) for this region: but these efforts are parallel to this study and are therefore not included here. We have added text to the discussion recommending the use of an ensemble of different ET and/or land surface models and we are working on this direction for exploring future implementations of our strategy.

We also added the following explanation regarding the choice of the resistance scheme to Section 3.4.1 (page 13, line 15): "The in-series scheme was selected as it has been demonstrated to estimate heat fluxes of densely vegetated areas better than the parallel or patch schemes (Kustas et al., 1999; Li et al., 2005; Colaizzi et al., 2012). In our study, the crops are not structured in rows, and at maturity the canopy covers the

entirety of the soil surface (with the exception of the beam tracks of the pivot), and so the area was considered as densely vegetated."

Anderson, M. C., Kustas, W. P., Norman, J. M., Hain, C. R., Mecikalski, J. R., Schultz, L., . . . Pimstein, A. (2011). Mapping daily evapotranspiration at field to continental scales using geostationary and polar orbiting satellite imagery.

Aragon, B., Malbeteau, Y., Fisher, J. B., McCabe, M. F. (2019). Evaluating the use of thermal imagery in crop water use management. Paper presented at the Geophysical Research Abstracts.

Colaizzi, P. D., Kustas, W. P., Anderson, M. C., Agam, N., Tolk, J. A., Evett, S. R., . . . O'Shaughnessy, S. A. (2012). Two-source energy balance model estimates of evapotranspiration using component and composite surface temperatures. Advances in Water Resources, 50, 134-151. doi:https://doi.org/10.1016/j.advwatres.2012.06.004

Kustas, W. P., Norman, J. M. (1999). Evaluation of soil and vegetation heat flux predictions using a simple two-source model with radiometric temperatures for partial canopy cover. Agricultural and Forest Meteorology, 94(1), 13-29. doi:https://doi.org/10.1016/S0168-1923(99)00005-2

Li, F., Kustas, W. P., Prueger, J. H., Neale, C. M. U., Jackson, T. J. (2005). Utility of Remote Sensing–Based Two-Source Energy Balance Model under Low- and High-Vegetation Cover Conditions. Journal of Hydrometeorology, 6(6), 878-891. doi:10.1175/jhm464.1

Nieto, H., Kustas, W. P., Torres-Rúa, A., Alfieri, J. G., Gao, F., Anderson, M. C., . . . McKee, L. G. (2019). Evaluation of TSEB turbulent fluxes using different methods for the retrieval of soil and canopy component temperatures from UAV thermal and multispectral imagery. Irrigation Science, 37(3), 389-406. doi:10.1007/s00271-018-0585-9

Zhuang, Q., Wu, B. (2015). Estimating Evapotranspiration from an Improved Two-

Source Energy Balance Model Using ASTER Satellite Imagery. Water, 7(12), 6673-6688.

**Comment 4**: The authors shows that the methodology is able to apply to large scale. While this is cool, I wonder if it is possible to evaluate the water budget by e.g., comparing with GRACE or other hydrological data? I'm still worried that water loss other than ET may not well captured by CABLE and may cause large uncertainty on the relationship between water input and ET.

**Author's response**: We thank the reviewer for this suggestion and we fully agree with the idea of water budget assessment in this region. In fact, this is an area of work that our research group has explored (Lopez et al., 2015; 2017) and this study forms part of an effort to address this. However, GRACE's ability to capture water storage changes is limited in terms of scale, with studies exploring only the largest (>200,000 $km^2$) basins and aquifers in the world (Famiglietti et al., 2014; Long et al., 2015; Richey et al., 2015). This limitation in scale was mentioned in the first paragraph of the introduction. However, we have now added the following text in the relevant paragraph in the Discussion section (page 20, line 22) to address this: "For example, the Al Jawf agricultural region as defined in this study (mapped agricultural area of about 2,500 $km^2$) is small compared to the scale of the Saq aquifer system that feeds it (about 500,000 $km^2$) and the recommended minimal size for GRACE studies (>200,000 $km^2$; Famiglietti et al., 2014; Long et al., 2015; Richey et al., 2015)."

We are currently working on estimating the water consumption within this entire region in order to compare the estimates with GRACE data.

Regarding water loss, within this environment the major loss occurs via evapotranspiration, as there are no perennial streams discharging to the sea and limited surface water-groundwater interaction (aquifers are very deep), making this region an ideal candidate to perform such an assessment. In more hydrologically complex environments, this approach would be more challenging to implement.

Famiglietti, J. S. (2014). The global groundwater crisis. Nature Climate Change, 4(11), 945-948.

Long, D., Longuevergne, L., Scanlon, B. R. (2015). Global analysis of approaches for deriving total water storage changes from GRACE satellites. Water Resources Research, 51(4), 2574-2594. doi:10.1002/2014wr016853

Lopez, O., McCabe, M. F. and Houborg, R. (2015). Evaluation of multiple satellite evaporation products in two dryland regions using GRACE. In Weber, T., McPhee, M.J. and Anderssen, R.S. (eds) MODSIM2015, 21st International Congress on Modelling and Simulation. Modelling and Simulation Society of Australia and New Zealand, December 2015, pp. 1379–1385. ISBN: 978-0-9872143-5-5. www.mssanz.org.au/modsim2015/F11/lopez.pdf

López, O., Houborg, R., McCabe, M. F. (2017). Evaluating the hydrological consistency of evaporation products using satellite-based gravity and rainfall data. Hydrology and Earth System Sciences, 21(1), 323-343.

Richey, A. S., Thomas, B. F., Lo, M.-H., Reager, J. T., Famiglietti, J. S., Voss, K., . . . Rodell, M. (2015). Quantifying renewable groundwater stress with GRACE. Water Resources Research, 51(7), 5217-5238. doi:10.1002/2015wr017349

Finally, we appreciate the reviewer's effort to provide specific comments within the manuscript that also help improve the description of our work. Below we include our responses to these:

**P3L14**: Landsat (7 8) is 8 days.

**Author's response**: In this work we only used Landsat 8 for the year 2015. We have edited the text in the manuscript to reflect this: "Landsat 8 data, on the other hand, has a spatial resolution of 30 m, allowing it to map individual fields with a revisit time of 16 days."

**P9L28**: Why not using Houborg's approach as I note he is in the author list. P9L30:

Need to specify the configuration of the training database derived from PROSAIL.

**Author's response**: The approach described in Houborg and McCabe (2018) works well when in situ data is available, which was the case for the smaller farm located near Riyadh. However, in this study we decided to undertake only the model-based approach, as no in situ leaf area index data for the Al Jawf region exists for the time period studied. We have added the following explanation to the manuscript to reflect this: "In our study, a coupled leaf-canopy model (PROSAIL) produced forward runs over a wide range of realizations, and these were used as a training dataset to develop estimates of LAI using a Random Forests (RF) approach. The inversion of the forward runs needed to derive LAI was based on the REGularized canopy reFLECtance model (REGFLEC; Houborg et al., 2015) which has been shown to be suitable for largely automated applications (Houborg and McCabe, 2016). The configuration of REGFLEC in this study was done as in Houborg and McCabe (2018), but we did not use the hybrid approach because of the lack of in situ LAI data in the larger region of Al Jawf."

Houborg, R., McCabe, M., Cescatti, A., Gao, F., Schull, M., Gitelson, A. (2015). Joint leaf chlorophyll content and leaf area index retrieval from Landsat data using a regularized model inversion system (REGFLEC). Remote Sensing of Environment, 159, 203-221.

Houborg, R., and McCabe, M. F. (2016). Adapting a regularized canopy reflectance model (REGFLEC) for the retrieval challenges of dryland agricultural systems. Remote Sensing of Environment, 186, 105-120. doi:https://doi.org/10.1016/j.rse.2016.08.017

Houborg, R., McCabe, M. F. (2018). A hybrid training approach for leaf area index estimation via Cubist and random forests machine-learning. IS-PRS Journal of Photogrammetry and Remote Sensing, 135, 173-188. doi:https://doi.org/10.1016/j.isprsjprs.2017.10.004

**P13L14**: Why series scheme is chosen as there are studies that recommend parallel scheme.

[Figure]

**Author's response**: The decision to use the in-series scheme was based on the type of crops during the study period and how they develop. For the most part, these crops are not highly structured in rows and at maturity, the canopy covers the entirety of the soil surface (except at the irrigation beam tracks) and so, the area can be considered as densely vegetated. Choosing the in-series approach allows to account for the coupling between soil and canopy (i.e., it allows for the interaction between soil and canopy heat fluxes). The in-series approach had better results than the parallel or patch scheme while looking at densely vegetated areas. For instance, Kustas et al. (1999) and Colaizzi et al. (2012) employed the in-series scheme in their studies with the former showing better performance than the parallel version of the model. We have now added the above description in the manuscript (page 13, line 15).

Kustas, W. P., Norman, J. M. (1999). Evaluation of soil and vegetation heat flux predictions using a simple two-source model with radiometric temperatures for partial canopy cover. Agricultural and Forest Meteorology, 94(1), 13-29. doi:https://doi.org/10.1016/S0168-1923(99)00005-2

Colaizzi, P. D., Kustas, W. P., Anderson, M. C., Agam, N., Tolk, J. A., Evett, S. R., . . . O'Shaughnessy, S. A. (2012). Two-source energy balance model estimates of evapotranspiration using component and composite surface temperatures. Advances in Water Resources, 50, 134-151. doi:https://doi.org/10.1016/j.advwatres.2012.06.004

**P13L19**: Is canopy height involved in the calculation? If so, how do you deal with that?

**Author's response**: We have now added this detail to the manuscript: "Given the lack of in-situ information for land cover and crop development stage, Hc could not be implemented as a linear function of NDVI. In this study, canopy height was prescribed to a constant value of 0.3 m."

**P13L26**: How do you calculate roughness length without sensible heat?

**Author's response**: The aerodynamic roughness length for heat and momentum

transport were defined to be as $z_{0M} = H_C/8$ and $z_{0H} = z_0M/exp\left(k_B^{-1}\right)$ where the $k_B^{-1}$ (i.e., $ln\left(z_{0M}/z_{0H}\right)$) parameter was set to 2 as in (Norman et al., 1995). We have now added this missing equation and explanation in the manuscript.

Norman, J. M., Kustas, W. P., Humes, K. S. (1995). Source approach for estimating soil and vegetation energy fluxes in observations of directional radiometric surface temperature. Agricultural and Forest Meteorology, 77(3), 263-293. doi:https://doi.org/10.1016/0168-1923(95)02265-Y

**P13L29**: Need a citation or justification. While shortwave radiation may follow Beer's Law (and there is impact of leaf angle distribution), longwave radiation is different (you may check CABLE's longwave radiation radiative transfer).

**Author's response**: We thank the reviewer for highlighting this error. We have now expanded and modified the text as follows:

The net radiation (Rn) is computed as:

$$R_n = (1 - \alpha) S_{dn} + L_{dn} - \varepsilon_{surf}\sigma_b LST^4$$

where $\alpha$ is the albedo, $S_{dn}$ and $L_{dn}$ are the incoming shortwave and longwave radiation components (derived from WRF data), $\varepsilon_{surf} = f_\varphi \varepsilon_{veg} + (1 - f_\varphi \varepsilon_{grd})$ is the surface emissivity with $f_\varphi$ described in Equation 10 and $\varepsilon_{veg} = 0.98$ and $\varepsilon_{grd} = 0.93$ are the canopy and soil emissivities respectively and $\sigma_b$ is the Stefan-Boltzmann constant.

The net radiation that reaches the canopy $R_{nc}$ is modelled as:

$$R_{nc} = R_n \left(1 - e^{-0.45LAI/\sqrt{2cos(\varphi_z)}}\right)$$

where $\varphi_z$ is the solar zenith angle. This simple parameterization for $R_{nc}$ was developed based on the Cupid model for dense canopies as described in Zhuang and Wu (2015).

Zhuang, Q., Wu, B. (2015). Estimating Evapotranspiration from an Improved Two-Source Energy Balance Model Using ASTER Satellite Imagery. Water, 7(12), 6673-6688.

**P14L8**: This assumes a spherical leaf angle distribution. Need elaboration.

**Author's response**: The following text was added to the manuscript after Equation 10: "The vegetation fraction assumes a spherical leaf angle distribution, which is a good approximation for general plant canopies, spreading the leaf area uniformly across solar zenith angles (Campbell 1998). For Saudi Arabia, at the Landsat overpass time, the extension coefficient $K_b$ = 0.5."

Campbell, G. S., Norman, J. (2012). An introduction to environmental biophysics: Springer Science Business Media.

**P14L9**: Then is there a sharp change when LAI=1 as you use two different models?

**Author's response**: Yes, there was a sharp change while using the OSEB rather than TSEB for low LAI values. The reason to implement OSEB for low LAI conditions was that in some instances TSEB showed unreasonably high bare soil evaporation rates, thereby leading to overestimation of water use in fallow/inactive fields. We have added the following text (in bold) to make this clear in the manuscript (page 14, line 9): "For pixels with low LAI values (LAI < 1) i.e. where the soil component is dominant, the canopy component was omitted by applying a simpler, one-source energy balance (OSEB). In the OSEB, the sensible heat flux is estimated by using a one-layer circuit network. Although this can lead to a sharp transition in ET for values around LAI=1, this was done to reduce the observed influence of unrealistic high bare soil evaporation values using TSEB, which would cause an overestimation of water use in fallow or inactive fields."

**P14L16**: Please specify parameter values.

**Author's response**: This refers to A and B in equation 12 (soil heat flux model of

Santanello and Friedl, 2003). The values of the A and B parameters were left as in the original parametrization as A = 0.31 and B = 74000 s. We have added this to the revised version.

Santanello, J. A. and Friedl, M.: Diurnal covariation in soil heat flux and net radiation, J. Appl. Meteorol., 42, 851–862, doi: 10.1175/1520-0450(2003)042<0851:DCISHF>2.0.CO;2, 2003.

**P14L20**: How is albedo calculated?

**Author's response**: We used the parameterization of Liang (2000). We have added this detail and reference to the manuscript.

Liang, S. (2001). Narrowband to broadband conversions of land surface albedo I: Algorithms. Remote Sensing of Environment, 76(2), 213-238. doi:https://doi.org/10.1016/S0034-4257(00)00205-4

**P14L27**: Isn't CABLE a dynamic vegetation model, i.e., LAI is calculated in a process? How could you use LAI as input which breaks the process?

Author's response: The version of CABLE that we used did not include a dynamic vegetation model. While there is a global coarse resolution monthly LAI dataset included as part of the auxiliary files (as described in Srbinovsky et al., 2013), we did not use this LAI product. Instead, we ingested LAI into the model based on our own estimates. This is now mentioned in the added text in section 3.4.2: "Finally, for leaf area index, we did not use the default MODIS-derived LAI, as the coarse resolution of this product is not sufficient to represent crop dynamics from individual fields. We used Landsat-derived LAI as described in Section 3.1"

Srbinovsky, J., Law, R., Pak, B. (2013). The Community Atmosphere Biosphere Land Exchange (CABLE) land surface model - User guide for CABLE-2.0. User guide.

**P15L5**: Is there a dynamic root consideration?

**Author's response**: To our knowledge, the version of CABLE we have used (2.3.4) does not feature a dynamic root module.

**P15L13**: How could you figure out these soil parameters?

**Author's response**: Please see the answer to the following question regarding CABLE parameters.

**P15L16**: I believe there are a large amount of parameters in CABLE which can influence ET response to water input. Do you do any calibration and sensitivity analysis?

**Author's response**: As noted in our earlier response, a specific calibration to CABLE parameters was not performed. Soil properties were based on the 1-degree global soil classification (Zobler, 1999) included as part of CABLE's auxiliary files for offline simulations, which in this region correspond to sandy loam soil (this detail has been now added to page 15, line 13). Given that the focus of the study is irrigated crops, it was important to set a dynamic leaf area index based on satellite observations – as described in Section 3.1.

Zobler, L. 1999. Global Soil Types, 1-Degree Grid (Zobler). Data set. Available on-line [http://www.daac.ornl.gov] from Oak Ridge National Laboratory Distributed Active Archive Center, Oak Ridge, Tennessee, U.S.A. doi:10.3334/ORNLDAAC/418.

Please also note the supplement to this comment:
https://www.hydrol-earth-syst-sci-discuss.net/hess-2020-50/hess-2020-50-AC1-supplement.pdf

**Supplement:**

[Figure]

**Figure S1: Comparison of monthly evaporation between the satellite-based estimates (TSEB) and hydrological model runs (CABLE) with irrigation estimated by the model inversion strategy. The line represents the monthly average over all agricultural fields in Al Jawf, while the shaded area shows ± the standard deviation.**

[Figure]

**Figure S2: Evaluation scatterplot of measured vs. estimated LE fluxes using the series-TSEB model approach over an irrigated maize field (Tawdeehiya farms) in 2016.**

---

## Author Comment (AC2) · 8 Jun 2020

We appreciate the reviewer comments and have attempted to improve the description of some of the methodological choices in the paper. In the following responses, we identify parts of the text that we added to address these comments.

**Comment 1**: Firstly, more details about CABLE model simulation should be given. For example, is there a crop model in it? How was the simulation conducted for different crops (maize-C4 and wheat-C3, and others)? What if there are more than two crop types (if that is possible in this region) within a single field? How was the spin-up conducted? What's the model's performance in simulating ET when giving observed

[Figure]

LAI compared with flux-tower or other ground-based measurements?

**Author's response**: Our aim was to independently approximate groundwater abstraction for the year 2015 of one of the largest agricultural regions in Saudi Arabia. We selected this year for operational reasons. Unfortunately, sufficient ground-truth data does not exist - and in fact, the motivation for the approach comes from the fact that there is a lack of any independent assessments to compare against. This study initiates the first efforts to provide an approximation of water consumption over a large region with sufficient granularity to compare values between individual fields, which will be refined once relevant ground-based data (slowly) become available (i.e. through on-farm metering) (page 22, lines 6-26).

To do this, we employ the CABLE land surface model (given its application in other dryland environments and as the land surface scheme for regional and global climate models; Zhang et al., 2009; Haverd et al., 2013; Hirsch et al., 2019): but the approach is not limited to any particular scheme. To provide further details on this approach, we have added additional information at the end of section 3.4.2:

"This version of CABLE is also available for offline global simulations using look-up tables for soil classification derived from Zobler (1999), and vegetation types defined by the International Geosphere and Biosphere Program (Loveland et al., 2000). CABLE includes monthly LAI data derived from MODIS data averaged from 2002 to 2009 (Gao et al., 2008; Ganguly et al., 2008), as specified in the CABLE user guide (Srbinovsky et al., 2013). As a first attempt, the default soil texture was used, and assumed as uniform across the studied region. However, LAI data was derived as described in section 3.1, as the coarse-scale MODIS-derived dataset is not representative of the actual crop growing patterns. The possibility of different crops and crop rotation in the same field within the year was considered, as explained in Section 3.3, using the clustering technique based on LAI data. One limitation of the framework is the lack of a crop identification module, which would improve the definition of vegetation characteristics. In this study, vegetation parameters were assigned based on the default CABLE cropland

vegetation class, as currently no crop identification strategy was implemented, other than the delineation and clustering technique."

We also added the following text referring to the spin-up of the model, which follows directly after the paragraph added above:

"Under basin-scale water budget studies, a spin-up of the model is generally required to achieve a realistic initial soil moisture state. This is normally done by running a representative year of meteorological data several times, or running several years prior to the start of the study period (Ajami et al., 2014; 2015), and assuming that the spin-up period is representative of the "normal" conditions. This assumption does not hold in our simulations because we are aiming to retrieve irrigated amounts, which could change from one year to the other, as different crops are grown. Therefore, this poses a challenge for how to represent the initial state of irrigated agricultural fields at the start of our simulations. In our study, the spin-up for each field was performed as follows: after estimation of the irrigation amount for one season, the model is run using this irrigation amount, and the final state is saved as the initial state for the next iterative process. However, the problem still lies with the spin-up of the first period. To solve this, we started by first running the groundwater abstraction strategy for a three-month period prior to the start of the study period, thus generating an initial state for the actual period of study."

Ajami, H., Evans, J. P., McCabe, M. F., Stisen, S. (2014). Technical Note: Reducing the spin-up time of integrated surface water–groundwater models. Hydrol. Earth Syst. Sci., 18(12), 5169-5179. doi:10.5194/hess-18-5169-2014

Ajami, H., McCabe, M. F., Evans, J. P. (2015). Impacts of model initialization on an integrated surface water–groundwater model. Hydrological Processes, 29(17), 3790-3801. doi:10.1002/hyp.10478

Ganguly, S., Samanta, A., Schull, M. A., Shabanov, N. V., Milesi, C., Nemani, R. R., . . . Myneni, R. B. (2008). Generating vegetation leaf area index Earth system data

record from multiple sensors. Part 2: Implementation, analysis and validation. Remote Sensing of Environment, 112(12), 4318-4332.

Gao, F., Morisette, J. T., Wolfe, R. E., Ederer, G., Pedelty, J., Masuoka, E., . . . Nightingale, J. (2008). An algorithm to produce temporally and spatially continuous MODIS-LAI time series. IEEE Geoscience and Remote Sensing Letters, 5(1), 60-64.

Haverd, V., Raupach, M., Briggs, P., Canadell, J., Isaac, P., Pickett-Heaps, C., . . . Wang, Z. (2013). Multiple observation types reduce uncertainty in Australia's terrestrial carbon and water cycles. Biogeosciences, 10(3), 2011.

Hirsch, A. L., Kala, J., Carouge, C. C., De Kauwe, M. G., Di Virgilio, G., Ukkola, A. M., . . . Abramowitz, G. (2019). Evaluation of the CABLEv2.3.4 Land Surface Model Coupled to NU-WRFv3.9.1.1 in Simulating Temperature and Precipitation Means and Extremes Over CORDEX AustralAsia Within a WRF Physics Ensemble. Journal of Advances in Modeling Earth Systems, 11(12), 4466-4488. doi:10.1029/2019ms001845

Loveland, T. R., Reed, B. C., Brown, J. F., Ohlen, D. O., Zhu, Z., Yang, L., Merchant, J. W. (2000). Development of a global land cover characteristics database and IGBP DISCover from 1 km AVHRR data. International Journal of Remote Sensing, 21(6-7), 1303-1330.

Srbinovsky, J., Law, R., Pak, B. (2013). The Community Atmosphere Biosphere Land Exchange (CABLE) land surface model - User guide for CABLE-2.0. User guide.

Zobler, L. 1999. Global Soil Types, 1-Degree Grid (Zobler). Data set. Available online [http://www.daac.ornl.gov] from Oak Ridge National Laboratory Distributed Active Archive Center, Oak Ridge, Tennessee, U.S.A. doi:10.3334/ORNLDAAC/418.

Zhang, L., Zhang, H., Li, Y. (2009). Surface energy, water and carbon cycle in China simulated by the Australian community land surface model (CABLE). Theoretical and Applied Climatology, 96(3), 375-394. doi:10.1007/s00704-008-0047-z

**Comment 2**: Secondly, the only validation reported in this manuscript is in Fig. 6

and 7 showing comparison between the annual/seasonal model estimation and farm reported groundwater abstraction. There is no validation on LAI and ET estimation from the satellite remote sensing. If there is uncertainties, how would they propagate to the final estimation of groundwater abstraction?

**Author's response**: We have previously addressed both the validation of LAI and ET data in the response to Reviewer 1, and thus here we include similar responses to these two issues: Houborg and McCabe (2018) described an approach for LAI estimation using a combination of physically-based estimates and in situ data. In their study, they showed reasonable LAI estimates for a small-scale (around 40 center-pivot fields) farm. In our study, as no comparable in situ data set exists for the Al Jawf region, we decided that a more physically-based approach was needed. That is why the PROSAIL implementation was explored. While not shown in the manuscript, we have implemented both approaches over the same region as in the Houborg and McCabe (2018) study and found that the second approach produced more reasonable LAI estimates. This provides confidence regarding the application of the second approach in this study. However, further work is needed to explore how the uncertainties in LAI (and other inputs) propagate to the final estimates of groundwater abstraction, a recommendation that we now added to the text (page 21, line 6) as follows: "The goal of this study was to provide a first approximation of regional groundwater abstraction independent from self-reported data, and for this, we have used a specific choice of models (i.e. TSEB and CABLE). Further investigation is required to determine the uncertainties of these models – as well as from other inputs such as LAI – and how they propagate through our groundwater abstraction framework. One approach that could help mitigate biases within specific models is to explore the use of multi-model estimates, which would also help provide ranges of groundwater abstraction."

Furthermore, the potential of TSEB for estimation of ET through remote sensing data has been well documented and validated over irrigated crops in semi-arid and arid regions (e.g. Colaizzi et al., 2012; Zhuang and Wu, 2015; Nieto et al., 2019). However,

validation of evaporative estimates using different remote sensing-based ET models (including TSEB) within the region examined in this study forms part of parallel efforts within our group (Aragon et al., 2019). We have now added a figure showing a comparison of estimated TSEB with in situ data for one of the irrigated fields in our study (Figure S2).

Aragon, B., Malbeteau, Y., Fisher, J. B., McCabe, M. F. (2019). Evaluating the use of thermal imagery in crop water use management. Paper presented at the Geophysical Research Abstracts.

Colaizzi, P. D., Kustas, W. P., Anderson, M. C., Agam, N., Tolk, J. A., Evett, S. R., . . . O'Shaughnessy, S. A. (2012). Two-source energy balance model estimates of evapotranspiration using component and composite surface temperatures. Advances in Water Resources, 50, 134-151. doi:https://doi.org/10.1016/j.advwatres.2012.06.004

Houborg, R., McCabe, M. F. (2018). A hybrid training approach for leaf area index estimation via Cubist and random forests machine-learning. IS-PRS Journal of Photogrammetry and Remote Sensing, 135, 173-188. doi:https://doi.org/10.1016/j.isprsjprs.2017.10.004

Nieto, H., Kustas, W. P., Torres-Rúa, A., Alfieri, J. G., Gao, F., Anderson, M. C., . . . McKee, L. G. (2019). Evaluation of TSEB turbulent fluxes using different methods for the retrieval of soil and canopy component temperatures from UAV thermal and multispectral imagery. Irrigation Science, 37(3), 389-406. doi:10.1007/s00271-018-0585-9

Zhuang, Q., Wu, B. (2015). Estimating Evapotranspiration from an Improved Two-Source Energy Balance Model Using ASTER Satellite Imagery. Water, 7(12), 6673-6688.

**Comment 3**: Thirdly, was the inverse modeling conducted at daily time scale? If it is, are we expecting irrigation every day, which is absolutely not true in the reality? The

authors reported the accumulated amount of irrigation water use at monthly and annual time scale. How about irrigation timing and times?

**Author's response**: Surprisingly, in Saudi Arabia, irrigation is indeed typically applied on a daily and continuous basis for prolonged periods during the crop growth cycle – hence the need to develop an approach that can address this major water use concern. The inverse modelling as applied in this work aimed at retrieving irrigation amounts at longer time scales. However, the frequency of satellite data used for 2015 is simply not sufficient to differentiate irrigation amounts between different crop developmental stages and seasons. Future development will aim at incorporating data form other platforms (Cubesats, Sentinel 2, etc) and determine whether sub-seasonal irrigation amounts can be obtained. However, we don't suspect that irrigation timing and times can be determined – although we are exploring other approaches that can attempt this, including CubeSats and Sentinel-1.

**Comment 4**: Fourthly, where did the authors assessed the model performance as described in section 3.5?

**Author's response**: This was done in Section 4.1 (Pivot-based framework performance at the Tawdeehiya Farm). As there is no ground-based data on irrigation applied at the Al Jawf region for the study period, we evaluated the model's performance using data from the smaller Tawdeehiya farm. As we described in the manuscript, this data presented some problems as well, but we obtained a Nash-Sutcliffe efficiency value of 0.38 and R squared value of 0.61 (Figure 6).

**Comment 5**: Finally, organization of this manuscript can be further improved. For example, Fig. 1 and Fig. 2 can be combined? Descriptions of TSEB and CABLE models can be simplified and details about them can be put into supplementary? Instead, the authors should focus more on simulation protocols there.

**Author's response**: We believe that the description of CABLE and TSEB were simplified in our manuscript relative to more detailed descriptions found in the literature

(Norman et al., 1995; Colaizzi et al., 2012; Wang et al., 2011; Kowalzcyk et al., 2013). However, based on comment 1, we added two paragraphs that add specific details of our strategy. We agree with the reviewer regarding the combination of figures 1 and 2, and we now have a combined figure (attached Figure 1).

**Figure 1 caption**: Figure 1: Location of the two study regions. Left: The Tawdeehiya farm in Al Kharj (southeast of Riyadh). A false color Landsat 8 image (2015/06/09) is shown to highlight active center-pivot fields over the desert environment. Right: The Al Jawf agricultural region in the north-west of Saudi Arabia spans two Landsat 8 tiles. Two false color images are shown: 2015/06/09 for path/row 172/39 (left) and 2015/06/19 for path/row 171/39 (right). Center-pivot fields are densely packed and largely uniform in size in the main area (30° N, 38.25° E), while in other areas they are sparser and less uniform (for example, the image on the right).

Colaizzi, P. D., Kustas, W. P., Anderson, M. C., Agam, N., Tolk, J. A., Evett, S. R., . . . O'Shaughnessy, S. A. (2012). Two-source energy balance model estimates of evapotranspiration using component and composite surface temperatures. Advances in Water Resources, 50, 134-151. doi:https://doi.org/10.1016/j.advwatres.2012.06.004

Kowalczyk, E., Stevens, L., Law, R. M., Dix, M., Wang, Y. P., Harman, I. N., Haynes, K., Srbinovsky, J., Pak, B., and Ziehn, T.: The land surface model component of ACCESS: description and impact on the simulated surface climatology, Aust. Meteorol. Ocean., 63, 65–82, 2013.

Norman, J. M., Kustas, W. P., Humes, K. S. (1995). Source approach for estimating soil and vegetation energy fluxes in observations of directional radiometric surface temperature. Agricultural and Forest Meteorology, 77(3), 263-293. doi:https://doi.org/10.1016/0168-1923(95)02265-Y

Wang, Y. P., Kowalczyk, E., Leuning, R., Abramowitz, G., Raupach, M. R., Pak, B., van Gorsel, E., and Luhar, A.: Diagnosing errors in a land surface model (CABLE) in the time and frequency domains, J. Geophys. Res., 116, G01034,

doi:10.1029/2010JG001385, 2011.

Please also note the supplement to this comment:
https://www.hydrol-earth-syst-sci-discuss.net/hess-2020-50/hess-2020-50-AC2-
supplement.pdf
* * *
[Figure]

[Figure]

**Fig. 1.** Revised Figure 1

**Supplement:**

[Figure]

**Figure S1: Comparison of monthly evaporation between the satellite-based estimates (TSEB) and hydrological model runs (CABLE) with irrigation estimated by the model inversion strategy. The line represents the monthly average over all agricultural fields in Al Jawf, while the shaded area shows ± the standard deviation.**

[Figure]

**Figure S2: Evaluation scatterplot of measured vs. estimated LE fluxes using the series-TSEB model approach over an irrigated maize field (Tawdeehiya farms) in 2016.**

---

## Author Response (AR1)

Dear Dr. Wang,

Thank you for your comments. We have revised the manuscript to address the reviewers' concerns as specified in our responses. The reviewers' comments helped us improve the description of the methodology and the discussion section. Specifically, we added two new paragraphs to the discussion section and added headings to improve its structure. In the methodology section, we added missing details both to the description of our approach and to the models used. Furthermore, we include two new supplementary figures (Fig. S1 and Fig. S2) related to the estimation of evaporation using TSEB and CABLE.

Below is a list of the specified changes, followed by point-by-point responses to the reviewers, and finally a marked-up version of the manuscript. We note that the response to reviewers' comments is the same as we have added in the Discussion version of the manuscript (HESSD). However, we have included the references cited in both responses in a single section following both responses.

**List of specified changes**

**Introduction**
- **Page 3, line 4**: added text to indicate that typical irrigation application in this region is on a daily basis.

**Description of study regions**
- **Figures 1 and 2**: we have combined these figures into one single figure. As a result, all Figures have now changed number (Figures 3 – 12 are now Figures 2 – 11).

**Description of the methodology (section 3)**
- **Page 8, lines 1 – 10**: We added text to this paragraph to improve the description of the interaction between the CABLE and TSEB models.
- **Page 9, line 31 – page 10, line 5**: We added text to 1) indicate the validation of LAI data in a previous study (Houborg and McCabe, 2018) and 2) add details regarding the use of the REGFLEC model in our study.
- **Section 3.4.1 (TSEB model description):** We revised this section substantially to add missing details such as the reason for using the in-series resistance scheme. We also corrected the error in the equations describing the net radiation reaching the canopy, which the first reviewer pointed out.
- **Section 3.4.2 (CABLE model description):** We revised this section substantially to add descriptions pertaining to the application of the CABLE model within our approach. As a result, among other added details, we added two new paragraphs at the end of this section.

**Discussion (section 5)**
- We added headings to the discussion section to provide a better structure. These include: **5.1)** Historical efforts towards quantifying agriculture water use, **5.2)** Demonstrating the

capability of the pivot-based groundwater abstraction estimation framework, and **5.3)** Limitations, potential extensions and applications.

- **Page 20, lines 16 – 27**: We added text to specify the scale required to provide a large-scale assessment of water fluxes, e.g. to validate using GRACE data.
- **Page 21, line 18**: We added a new paragraph to emphasize the need for this type of studies in data-limited regions, and the fact that our results do reflect expectations compared with the limited independent data sets.
- **Page 21, lines 18 – 25**: We moved this paragraph to section 5.3.
- **Section 5.3**: We added a new paragraph to emphasize that our approach is not limited by the choice of TSEB and CABLE, and to recommend a further direction to explore other models (e.g. ALEXI/DisALEXI).

**References**
- We added new references that we cited in the added text.

**Author's response to reviewer 1 (R1):**

We thank the reviewer for their comments and questions that have helped to improve the description of the methodology. As we integrate a large number of different elements in this framework, we recognize that some descriptions can be improved for clarity.

> **Comment 1:** The land surface model CABLE is the core of the methodology. Only if CABLE could correctly simulate each hydrological component, the water input could be inferred from ET. This is a big concern as land surface models are known to overestimate the coupling between soil moisture and ET ([10.1029/2018WR023469](10.1029/2018WR023469)).

**Author's response**: We agree that there are challenges in hydrological component estimation with land surface (and even process) models. However, models serve as the best tool for providing an interpretive framework, particularly when used in the context of ingesting or evaluating against available observations. While a particular combination of models were used in the current approach (CABLE and TSEB), based on both past model experiences and their previously evaluated suitability, we do explicitly recommend exploring the use of different models, or even an ensemble of models. Our framework certainly provides this flexibility. To highlight this, we have now included the following text in the Discussion section (page 21, line 6): "The goal of this study was to provide a first approximation of regional groundwater abstraction independent from self-reported data, and for this, we have used a specific combination of models (i.e. TSEB and CABLE). Further investigation is required to determine the uncertainties of these models – as well as other inputs – and how they might propagate through our groundwater abstraction framework. One approach that could help mitigate biases within specific models is to explore the use of multi-model estimates, which would also help provide estimated ranges of groundwater abstraction."

**Comment 2:** How the model is calibrated in the study area? What's the sensitivity of ET to water input? What's the model performance in forward simulation? What's the underlying uncertainty of model inversion? To what degree TSEB ET improve CABLE ET? All these questions are not addressed and therefore the methodology is not convincing, even though the final results look good.

**Author's response**: We have split our response to answer these questions below. However, as a general response, it is important to note that this study represents a demonstration approach to provide meaningful information in a data poor environment. That is, it was not intended as a calibration/validation effort of a predictive model (or integration of models) framework, but rather as an application that builds on decades of research in the use of remote sensing data and land surface models.

**Comment 2.1:** How is the model calibrated in the study area?
**Author's response**: We did not perform a traditional calibration of parameters in the CABLE model. One reason is that this study is not performed within a defined basin with an observable runoff component, which is the usual approach to model calibration. Absent such observations, we are limited to running the model in default mode. While there have been efforts to undertake multi-observation based calibration approaches (which our team has explored previously; McCabe et al., 2005; Stisen et al., 2011), these are certainly not commonplace: and again run into the problem that one of our observations is actually used in the optimization target (i.e. independently observed ET). Our study "area" is in fact 5000+ individually delineated center-pivot fields, in which we employ the model as a 1D representation to capture vertical moisture transport. That is, we are most interested in using the model physics to capture the underlying hydrological processes related to hydrological partitioning. Therefore, CABLE parameterization was done based on regional soil parameters but with vegetation characteristics determined by satellite observations (e.g. LAI). Using these parameters, we employ the model as an optimization tool to infer the irrigation amount. Even running the model in a "forward" mode is not realistic, as rainfall is non-existent – and without a coupled irrigation module, we developed an approach to infer this variable indirectly.

**Comment 2.2:** What's the sensitivity of ET to water input and what's the model performance in forward simulation? To what degree TSEB ET improve CABLE ET?

**Author's response**:  It is important to note that a forward simulation of the land surface model cannot be performed, precisely because the irrigation amount is not known (and rainfall is non-existent). That is, implementing the land surface model using only the limited rainfall amount in this region, leads to an almost complete lack of evaporation signal. On the other hand, TSEB ET is derived from satellite observations, and does not require prescribing the irrigation amount applied. Therefore, TSEB ET is used to infer the irrigation amount that would be needed in CABLE to reproduce the estimated ET amount. In that sense, TSEB ET improves the estimation of CABLE ET by providing the missing irrigation component in this region.

**Comment 2.3:** What's the underlying uncertainty of model inversion?

**Author's response**: In this and many other such regions, ground based collections of irrigation data do not exist for individual fields at a sufficiently large scale to evaluate the uncertainty of model inversion. These and other issues were mentioned in Section 4.1. Ultimately, our study is a demonstration of using the best available data and tools to undertake an analysis in a data-limited region. This is very much a developing world problem, but there are numerous "developed" world cases where data to inform model set-up and evaluation is absent. Using the strategy proposed here, we were able to provide results that reflect expectations when compared to the limited available datasets (both at farm and regional scale), i.e. a positive outcome given the scenario where independent data is not available to inform decision makers.

> **Comment 3:** ET is the link between the land surface model and satellite remote sensing. However, no ET result is shown in the manuscript at all, and there's no comparison between TSEB ET, CABLE ET and field observation. While TSEB is a reasonable model, it is sensitive to the accuracy of LST but LST retrieval is usually error-prone (that's why ALEXI is developed to mitigate this issue), in particular for the TIRS onboard Landsat. TSEB is also sensitive to some configurations such as the selection of resistance scheme. The authors use "in-series" scheme while other papers use "in-parallel" scheme. How these intermediate steps were determined and how they perform worth demonstration.

**Author's response**: As the main contribution of this work is to estimate groundwater abstraction, we focused our attention only on this component in the results section. However, we have now added a figure showing ET results to add confidence to the results as supplementary data (Figure S2).

We would note that the potential of TSEB (and other remote sensing models) for estimation of ET over irrigated crops in semi-arid and arid regions has been well documented and validated (e.g. Colaizzi et al., 2012; Zhuang and Wu, 2015; Nieto et al., 2019), including initial efforts from our group to validate TSEB (among other evaporation models) within Saudi Arabia (Aragon et al., 2019). At some point (and after decades of research), the hope is that such remote sensing based approaches can ultimately move beyond being purely research tools, to actually be implemented to provide guidance for operational management. That is an aspect that we explore here.

However, to address the concern, we have also added a figure (Figure S1) showing a comparison of TSEB and measured LE over an irrigated field in one of the study regions (Tawdeehiya farm; see Figure 1). TSEB was chosen to reduce the already high computational load of running CABLE and an ET model over this large region. Nevertheless, we are aware of the sensitivity of TSEB to the accuracy to the LST retrieval, and we are actively exploring the use of ALEXI and its associated disaggregation scheme (DisALEXI; Anderson et al., 2011) for this region: but these efforts are parallel to this study and are therefore not included here. We have added text to the discussion recommending the use of an ensemble of different ET and/or land surface models and we are working on this direction for exploring future implementations of our strategy.

We also added the following explanation regarding the choice of the resistance scheme to Section 3.4.1 (page 13, line 15): "The in-series scheme was selected as it has been demonstrated to estimate heat fluxes of densely vegetated areas better than the parallel or patch schemes (Kustas et al., 1999; Li et al., 2005; Colaizzi et al., 2012). In our study, the crops are not structured in rows, and at maturity the canopy covers the entirety of the soil surface (with the exception of the beam tracks of the pivot), and so the area was considered as densely vegetated."

**Comment 4**: The authors shows that the methodology is able to apply to large scale. While this is cool, I wonder if it is possible to evaluate the water budget by e.g., comparing with GRACE or other hydrological data? I'm still worried that water loss other than ET may not well captured by CABLE and may cause large uncertainty on the relationship between water input and ET.

**Author's response**: We thank the reviewer for this suggestion and we fully agree with the idea of water budget assessment in this region. In fact, this is an area of work that our research group has explored (Lopez et al., 2015; 2017) and this study forms part of an effort to address this. However, GRACE's ability to capture water storage changes is limited in terms of scale, with studies exploring only the largest (>200,000 $km^2$) basins and aquifers in the world (Famiglietti et al., 2014; Long et al., 2015; Richey et al., 2015). This limitation in scale was mentioned in the first paragraph of the introduction. However, we have now added the following text in the relevant paragraph in the Discussion section (page 20, line 22) to address this: "For example, the Al Jawf agricultural region as defined in this study (mapped agricultural area of about 2,500 $km^2$) is small compared to the scale of the Saq aquifer system that feeds it (about 500,000 $km^2$) and the recommended minimal size for GRACE studies (>200,000 $km^2$; Famiglietti et al., 2014; Long et al., 2015; Richey et al., 2015)."

We are currently working on estimating the water consumption within this entire region in order to compare the estimates with GRACE data.

Regarding water loss, within this environment the major loss occurs via evapotranspiration, as there are no perennial streams discharging to the sea and limited surface water-groundwater interaction (aquifers are very deep), making this region an ideal candidate to perform such an assessment. In more hydrologically complex environments, this approach would be more challenging to implement.

Finally, we appreciate the reviewer's effort to provide specific comments within the manuscript that also help improve the description of our work. Below we include our responses to these:

**P3L14**: Landsat (7 & 8) is 8 days.

**Author's response**: In this work we only used Landsat 8 for the year 2015. We have edited the text in the manuscript to reflect this: "Landsat 8 data, on the other hand, has a spatial resolution of 30 m, allowing it to map individual fields with a revisit time of 16 days."

**P9L28**: Why not using Houborg's approach as I note he is in the author list. **P9L30**: Need to specify the configuration of the training database derived from PROSAIL.

**Author's response**: The approach described in Houborg and McCabe (2018) works well when in situ data is available, which was the case for the smaller farm located near Riyadh. However, in this study we decided to undertake only the model-based approach, as no in situ leaf area index data for the Al Jawf region exists for the time period studied. We have added the following text (**bold**) to the manuscript to reflect this: "In our study, a coupled leaf-canopy model (PROSAIL) produced forward runs over a wide range of realizations, and these were used as a training dataset to develop estimates of LAI using a Random Forests (RF) approach. **The inversion of the forward runs needed to derive LAI was based on the REGularized canopy reFLECtance model (REGFLEC; Houborg et al., 2015) which has been shown to be suitable for largely automated applications (Houborg and McCabe, 2016). The configuration of REGFLEC in this study was done as in Houborg and McCabe (2018a), but we did not use the hybrid approach because of the lack of in situ LAI data in the larger region of Al Jawf.**"

**P13L14**: Why series scheme is chosen as there are studies that recommend parallel scheme.

**Author's response**: The decision to use the in-series scheme was based on the type of crops during the study period and how they develop. For the most part, these crops are not highly structured in rows and at maturity, the canopy covers the entirety of the soil surface (except at the irrigation beam tracks) and so, the area can be considered as densely vegetated. Choosing the in-series approach allows to account for the coupling between soil and canopy (i.e., it allows for the interaction between soil and canopy heat fluxes). The in-series approach had better results than the parallel or patch scheme while looking at densely vegetated areas. For instance, Kustas et al. (1999) and Colaizzi et al. (2012) employed the in-series scheme in their studies with the former showing better performance than the parallel version of the model. We have now added the above description in the manuscript (page 13, line 15).

**P13L19**: Is canopy height involved in the calculation? If so, how do you deal with that?

**Author's response**: We have now added this detail to the manuscript: "Given the lack of in-situ information for land cover and crop development stage, Hc could not be implemented as a linear function of NDVI. In this study, canopy height was prescribed to a constant value of 0.3 m."

**P13L26**: How do you calculate roughness length without sensible heat?

**Author's response**: The aerodynamic roughness length for heat and momentum transport were defined to be as $z_{0M} = H_C/8$ and $z_{0H} = z_{0M}/\exp(k_B{}^{-1})$ where the $k_B{}^{-1}$ (i.e., $\ln(z_{0M}/z_{0H})$ ) parameter was set to 2 as in (Norman et al., 1995). We have now added this missing equation and explanation in the manuscript.

**P13L29**: Need a citation or justification. While shortwave radiation may follow Beer's Law (and there is impact of leaf angle distribution), longwave radiation is different (you may check CABLE's longwave radiation radiative transfer).

**Author's response**: We thank the reviewer for highlighting this error. We have now expanded and modified the text as follows:

The net radiation ($R_n$) is computed as:
$$R_n = (1 - \alpha)S_{dn} + L_{dn} - \varepsilon_{surf}\sigma_b LST^4$$
where $\alpha$ is the albedo, $S_{dn}$ and $L_{dn}$ are the incoming shortwave and longwave radiation components (derived from WRF data), $\varepsilon_{surf} = (f_\varphi \varepsilon_{veg} + (1 - f_\varphi)\varepsilon_{grd})$ is the surface emissivity with $f_\varphi$ described in Equation 10 and $\varepsilon_{veg} = 0.98$ and $\varepsilon_{grd} = 0.93$ are the canopy and soil emissivities respectively and $\sigma_b$ is the Stefan–Boltzmann constant.

The net radiation that reaches the canopy Rnc is modelled as:
$$R_{nc} = R_n \left(1 - e^{-0.45LAI/\sqrt{2\cos(\varphi_z)}}\right)$$
Where $\varphi_z$ is the solar zenith angle. This simple parameterization for $R_{nc}$ was developed based on the Cupid model for dense canopies as described in Zhuang and Wu (2015).

**P14L8**: This assumes a spherical leaf angle distribution. Need elaboration.

**Author's response**: The following text was added to the manuscript after Equation 10: "The vegetation fraction assumes a spherical leaf angle distribution, which is a good approximation for general plant canopies, spreading the leaf area uniformly across solar zenith angles (Campbell 1998). For Saudi Arabia, at the Landsat overpass time, the extension coefficient Kb =~0.5."

**P14L9**: Then is there a sharp change when LAI=1 as you use two different models?

**Author's response**: Yes, there was a sharp change while using the OSEB rather than TSEB for low LAI values. The reason to implement OSEB for low LAI conditions was that in some instances TSEB showed unreasonably high bare soil evaporation rates, thereby leading to overestimation of water use in fallow/inactive fields. We have added the following text (in bold) to make this clear in the manuscript (page 14, line 9): "For pixels with low LAI values (LAI < 1) i.e. where the soil component is dominant, the canopy component was omitted by applying a simpler, one-source energy balance (OSEB). In the OSEB, the sensible heat flux is estimated by using a one-layer circuit network. **Although this can lead to a sharp transition in ET for values around LAI=1, this was done to reduce the observed influence of unrealistic high bare soil evaporation values using TSEB, which would cause an overestimation of water use in fallow or inactive fields.**"

**P14L16**: Please specify parameter values.

**Author's response**: This refers to A and B in equation 12 (soil heat flux model of Santanello and Friedl, 2003). The values of the A and B parameters were left as in the original parametrization as A = 0.31 and B = 74000 s. We have added this to the revised version.

P14L20: How is albedo calculated?

**Author's response**: We used the parameterization of Liang (2000). We have added this detail and reference to the manuscript.

P14L27: Isn't CABLE a dynamic vegetation model, i.e., LAI is calculated in a process? How could you use LAI as input which breaks the process?

**Author's response**: The version of CABLE that we used did not include a dynamic vegetation model. While there is a global coarse resolution monthly LAI dataset included as part of the auxiliary files (as described in Srbinovsky et al., 2013), we did not use this LAI product. Instead, we ingested LAI into the model based on our own estimates. This is now mentioned in the added text in section 3.4.2: "Finally, for leaf area index, we did not use the default MODIS-derived LAI, as the coarse resolution of this product is not sufficient to represent crop dynamics from individual fields. We used Landsat-derived LAI as described in Section 3.1"

P15L5: Is there a dynamic root consideration?

**Author's response**: To our knowledge, the version of CABLE we have used (2.3.4) does not feature a dynamic root module.

P15L13: How could you figure out these soil parameters?

**Author's response**: Please see the answer to the following question regarding CABLE parameters.

P15L16: I believe there are a large amount of parameters in CABLE which can influence ET response to water input. Do you do any calibration and sensitivity analysis?

**Author's response**: As noted in our earlier response, a specific calibration to CABLE parameters was not performed. Soil properties were based on the 1-degree global soil classification (Zobler, 1999) included as part of CABLE's auxiliary files for offline simulations, which in this region correspond to sandy loam soil (this detail has been now added to page 15, line 13). Given that the focus of the study is irrigated crops, it was important to set a dynamic leaf area index based on satellite observations – as described in Section 3.1.

**Author's response to reviewer 2 (R2):**

This manuscript presented an interesting framework to estimate groundwater abstractions for agricultural irrigation. The framework is novel as it integrate land surface modeling, satellite remote sensing, and inverse modeling. This work was built upon several other works from the same group of people. The manuscript is well written, though the organization part can be further improved. I have the following concerns for the authors to consider:

**Author's response**:
We appreciate the reviewer comments and have attempted to improve the description of some of the methodological choices in the paper. In the following responses, we identify parts of the text that we added to address these comments.

**Comment 1:** Firstly, more details about CABLE model simulation should be given. For example, is there a crop model in it? How was the simulation conducted for different crops (maize-C4 and wheat-C3, and others)? What if there are more than two crop types (if that is possible in this region) within a single field? How was the spin-up conducted? What's the model's performance in simulating ET when giving observed LAI compared with flux-tower or other ground-based measurements?

**Author's response**: Our aim was to independently approximate groundwater abstraction for the year 2015 of one of the largest agricultural regions in Saudi Arabia. We selected this year for operational reasons. Unfortunately, sufficient ground-truth data does not exist - and in fact, the motivation for the approach comes from the fact that there is a lack of any independent assessments to compare against. This study initiates the first efforts to provide an approximation of water consumption over a large region with sufficient granularity to compare values between individual fields, which will be refined once relevant ground-based data (slowly) become available (i.e. through on-farm metering) (page 22, lines 6-26).

To do this, we employ the CABLE land surface model (given its application in other dryland environments and as the land surface scheme for regional and global climate models; Zhang et al., 2009; Haverd et al., 2013; Hirsch et al., 2019): but the approach is not limited to any particular scheme. To provide further details on this approach, we have added additional information at the end of section 3.4.2:

"This version of CABLE is also available for offline global simulations using look-up tables for soil classification derived from Zobler (1999), and vegetation types defined by the International Geosphere and Biosphere Program (Loveland et al., 2000). CABLE includes monthly LAI data derived from MODIS data averaged from 2002 to 2009 (Gau et al., 2008; Ganguly et al., 2008), as specified in the CABLE user guide (Srbinovsky et al., 2013). As a first attempt, the default soil texture was used, and assumed as uniform across the studied region. However, LAI data was derived as described in section 3.1, as the coarse-scale MODIS-derived dataset is not representative of the actual crop growing patterns. The possibility of different crops and crop

rotation in the same field within the year was considered, as explained in Section 3.3, using the clustering technique based on LAI data. One limitation of the framework is the lack of a crop identification module, which would improve the definition of vegetation characteristics. In this study, vegetation parameters were assigned based on the default CABLE cropland vegetation class, as currently no crop identification strategy was implemented, other than the delineation and clustering technique."

We also added the following text referring to the spin-up of the model, which follows directly after the paragraph added above:

"Under basin-scale water budget studies, a spin-up of the model is generally required to achieve a realistic initial soil moisture state. This is normally done by running a representative year of meteorological data several times, or running several years prior to the start of the study period (Ajami et al., 2014; 2015), and assuming that the spin-up period is representative of the "normal" conditions. This assumption does not hold in our simulations because we are aiming to retrieve irrigated amounts, which could change from one year to the other, as different crops are grown. Therefore, this poses a challenge for how to represent the initial state of irrigated agricultural fields at the start of our simulations. In our study, the spin-up for each field was performed as follows: after estimation of the irrigation amount for one season, the model is run using this irrigation amount, and the final state is saved as the initial state for the next iterative process. However, the problem still lies with the spin-up of the first period. To solve this, we started by first running the groundwater abstraction strategy for a three-month period prior to the start of the study period, thus generating an initial state for the actual period of study."

> **Comment 2**: Secondly, the only validation reported in this manuscript is in Fig. 6 and 7 showing comparison between the annual/seasonal model estimation and farm reported groundwater abstraction. There is no validation on LAI and ET estimation from the satellite remote sensing. If there is uncertainties, how would they propagate to the final estimation of groundwater abstraction?

**Author's response**: We have previously addressed both the validation of LAI and ET data in the response to Reviewer 1, and thus here we include similar responses to these two issues:
Houborg and McCabe (2018) described an approach for LAI estimation using a combination of physically-based estimates and in situ data. In their study, they showed reasonable LAI estimates for a small-scale (around 40 center-pivot fields) farm. In our study, as no comparable in situ data set exists for the Al Jawf region, we decided that a more physically-based approach was needed. That is why the PROSAIL implementation was explored. While not shown in the manuscript, we have implemented both approaches over the same region as in the Houborg and McCabe (2018) study and found that the second approach produced more reasonable LAI estimates. This provides confidence regarding the application of the second approach in this study. However, further work is needed to explore how the uncertainties in LAI (and other inputs) propagate to the final estimates of groundwater abstraction, a recommendation that we now added to the text (page 21, line 6) as follows: "The goal of this study was to provide a first approximation of regional groundwater abstraction independent from self-reported data, and for this, we have used a

specific choice of models (i.e. TSEB and CABLE). Further investigation is required to determine the uncertainties of these models – as well as from other inputs such as LAI – and how they propagate through our groundwater abstraction framework. One approach that could help mitigate biases within specific models is to explore the use of multi-model estimates, which would also help provide ranges of groundwater abstraction."

Furthermore, the potential of TSEB for estimation of ET through remote sensing data has been well documented and validated over irrigated crops in semi-arid and arid regions (e.g. Colaizzi et al., 2012; Zhuang and Wu, 2015; Nieto et al., 2019). However, validation of evaporative estimates using different remote sensing-based ET models (including TSEB) within the region examined in this study forms part of parallel efforts within our group (Aragon et al., 2019). We have now added a figure showing a comparison of estimated TSEB with in situ data for one of the irrigated fields in our study (Figure S1).

> **Comment 3:** Thirdly, was the inverse modeling conducted at daily time scale? If it is, are we expecting irrigation every day, which is absolutely not true in the reality? The authors reported the accumulated amount of irrigation water use at monthly and annual time scale. How about irrigation timing and times?

**Author's response**: Surprisingly, in Saudi Arabia, irrigation is indeed typically applied on a daily and continuous basis for prolonged periods during the crop growth cycle – hence the need to develop an approach that can address this major water use concern. The inverse modelling as applied in this work aimed at retrieving irrigation amounts at longer time scales. However, the frequency of satellite data used for 2015 is simply not sufficient to differentiate irrigation amounts between different crop developmental stages and seasons. Future development will aim at incorporating data form other platforms (Cubesats, Sentinel 2, etc) and determine whether sub-seasonal irrigation amounts can be obtained. However, we don't suspect that irrigation timing and times can be determined – although we are exploring other approaches that can attempt this, including CubeSats and Sentinel-1.

> **Comment 4:** Fourthly, where did the authors assessed the model performance as described in section 3.5?

This was done in Section 4.1 (Pivot-based framework performance at the Tawdeehiya Farm). As there is no ground-based data on irrigation applied at the Al Jawf region for the study period, we evaluated the model's performance using data from the smaller Tawdeehiya farm. As we described in the manuscript, this data presented some problems as well, but we obtained a Nash-Sutcliffe efficiency value of 0.38 and R squared value of 0.61 (Figure 6).

> **Comment 5:** Finally, organization of this manuscript can be further improved. For example, Fig. 1 and Fig. 2 can be combined? Descriptions of TSEB and CABLE models can be simplified and details about them can be put into supplementary? Instead, the authors should focus more on simulation protocols there.

We believe that the description of CABLE and TSEB were simplified in our manuscript relative to more detailed descriptions found in the literature (Norman et al., 1995; Colaizzi et al., 2012; Wang et al., 2011; Kowalzcyk et al., 2013). However, based on comment 1, we added two paragraphs that add specific details of our strategy. We agree with the reviewer regarding the combination of figures 1 and 2, and we now have combined these into one single figure.

[revised manuscript text omitted]